# Unifying Causal Representation Learning with the Invariance Principle

**Dingling Yao**, **Dario Rancati**, **Riccardo Cadei**, **Marco Fumero**, and **Francesco Locatello**

Institute of Science and Technology Austria

**Editors:** Marco Fumero, Clementine Domine, Zorah Lähner, Donato Crisostomi, Luca Moschella, Kimberly Stachenfeld

## Abstract

Causal representation learning aims at recovering latent causal variables from high-dimensional observations to solve causal downstream tasks, such as predicting the effect of new interventions or more robust classification. A plethora of methods have been developed, each tackling carefully crafted problem settings that lead to different types of identifiability. The folklore is that these different settings are important, as they are often linked to different rungs of Pearl's causal hierarchy, although not all neatly fit. Our main contribution is to show that *many existing causal representation learning approaches methodologically align the representation to known data symmetries*. Identification of the variables is guided by equivalence classes across different "data pockets" that are not necessarily causal. This result suggests important implications, allowing us to unify many existing approaches in a single method that can mix and match different assumptions, including non-causal ones, based on the invariances relevant to our application. It also significantly benefits applicability, which we demonstrate by improving treatment effect estimation on real-world high-dimensional ecological data. Overall, this paper clarifies the role of causality assumptions in the discovery of causal variables and shifts the focus to preserving data symmetries.

## 1 Introduction

Causal representation learning (CRL) [1] posits that many real-world high-dimensional perceptual data can be described through a simplified latent structure specified by a few interpretable low-dimensional causally-related variables. Many existing approaches in causal representation learning carefully formulate their problem settings to guarantee identifiability [2–7]. However, some CRL works may not perfectly fit within this causal language framework; for instance, the problem setting of temporal CRL works [8–11] ,domain generalization [12–14] and certain multi-task learning approaches [15, 16] are sometimes framed as informally related to causal representation learning. This has resulted in a variety of methods and findings, some of which rely on assumptions that are not always tailored for practical, real-world applications [17]. This paper contributes a unified rephrasing of many existing nonparametric CRL works through the lens of invariance. We observe that *many existing causal representation approaches share methodological similarities, particularly in aligning the representation with known data symmetries*, while differing primarily in how the invariance principle is invoked. We highlight our contributions as follows:

- We propose a unified rephrasing for existing nonparametric CRL approaches leveraging the invariance principles and prove latent variable identifiability in this general setting (§ 3). We show that 31 existing identification results can be seen as special cases directly implied by our framework (Tab. 2).

Proceedings of the II edition of the Workshop on Unifying Representations in Neural Models (UniReps 2024).

- We formalize different definitions of "identifiability" highlighting their connections, and demonstrating how they can be addressed within our framework (App. C.1)
- We analyze the case of partial graph identification, drawing a distinction between the causal assumptions necessary for graph discovery and those required for variable discovery (App. C.2).
- Our framework is broadly applicable across a range of practical settings, with improved results on real-world experimental ecology data using the causal inference benchmark of [17] (§ 4). We demonstrate that existing methods only require a form of distributional invariance for identification, without needing access to interventions (App. F).

## 2  Problem Setting

This section defines our problem setting using standard CRL concepts and assumptions (Formal definitions are deferred to App. B, a comprehensive summary of notations is provided in App. A). While prior works in CRL typically categorize their settings using established causal language (e.g. 'counterfactual,' 'interventional,' or 'observational'), our approach introduces a more general invariance principle that aims to unify diverse problem settings. We introduce the following concepts as mathematical tools to describe our data generating process.

**Definition 2.1** (Invariance property)**.** Let $A \subseteq [N]$ be an index subset of the Euclidean space $\mathbb{R}^N$ and let $\sim_\iota$ be an equivalence relationship on $\mathbb{R}^{|A|}$, with $A$ of known dimension. Let $\mathcal{M} := \mathbb{R}^{|A|} / \sim_\iota$ be the quotient of $\mathbb{R}^{|A|}$ under this equivalence relationship; $\mathcal{M}$ is a topological space equipped with the quotient topology. Let $\iota : \mathbb{R}^{|A|} \to \mathcal{M}$ be the projection onto the quotient induced by the equivalence relationship $\sim_\iota$. This projection $\iota$ is termed the *invariant property* of this equivalence relation. Two vectors $\mathbf{a}, \mathbf{b} \in \mathbb{R}^{|A|}$ are invariant under $\iota$ if and only if they belong to the same $\sim_\iota$ equivalence class, i.e.: $\iota(\mathbf{a}) = \iota(\mathbf{b}) \Leftrightarrow \mathbf{a} \sim_\iota \mathbf{b}$. and $\iota(\mathbf{z}_A) = \iota(\tilde{\mathbf{z}}_A) \Leftrightarrow \mathbf{z}_A \sim_\iota \tilde{\mathbf{z}}_A$.

Extending this definition to the whole latent space $\mathbb{R}^N$, a pair of latents $\mathbf{z}, \tilde{\mathbf{z}} \in \mathbb{R}^N$ are ***non-trivially invariant on a subset*** $A \subseteq [N]$ ***under the property*** $\iota$ only if (i) the invariance property $\iota$ holds on the indices $A \subseteq [N]$ in the sense that $\iota(\mathbf{z}_A) = \iota(\tilde{\mathbf{z}}_A)$; (ii) for any smooth function $h_1, h_2 : \mathbb{R}^N \to \mathbb{R}^{|A|}$, the invariance property between $\mathbf{z}, \tilde{\mathbf{z}}$ *breaks* under the $h_1, h_2$ transformations if $h_1$ or $h_2$ *directly* depends on some other component $\mathbf{z}_q$ with $q \in [N] \setminus A$. Taking $h_1$ and $\mathbf{z}$ as an example, we have:

$$\exists q \in [N] \setminus A, \mathbf{z}^* \in \mathbb{R}^N, \quad s.t. \; \frac{\partial h_1}{\partial \mathbf{z}_q}(\mathbf{z}^*) \text{ exists and is non zero} \quad \Rightarrow \quad \iota(h_1(\mathbf{z})) \neq \iota(h_2(\tilde{\mathbf{z}}))$$

i.e. given that the partial derivative of $h_1$ w.r.t. some latent variable $\mathbf{z}_q \in \mathbf{z}_{[N] \setminus A}$ is non-zero at some point $\mathbf{z}^* \in \mathbb{R}^N$, $h_1(\mathbf{z}), h_2(\mathbf{z})$ violates invariance principle in the sense that $\iota(h_1(\mathbf{z})) \neq \iota(h_2(\tilde{\mathbf{z}}))$.

We denote by $\mathcal{S}_{\mathbf{z}} := \{\mathbf{z}^1, \ldots, \mathbf{z}^K\}$ the set of latent random vectors with $\mathbf{z}^k \in \mathbb{R}^N$ and write its joint distribution as $P_{\mathcal{S}_{\mathbf{z}}}$. The joint distribution $P_{\mathcal{S}_{\mathbf{z}}}$ has a probability density $p_{\mathcal{S}_{\mathbf{z}}}(z^1, \ldots, z^K)$. Each individual random vector $\mathbf{z}^k \in \mathcal{S}_{\mathbf{z}}$ follows the marginal density $p_{\mathbf{z}^k}$ with the non-degenerate support $\mathcal{Z}^k \subseteq \mathbb{R}^N$, whose interion is a non-empty open set of $\mathbb{R}^N$.

**Definition 2.2** (Observable of a set of latent random vectors)**.** Consider a set of random vectors $\mathcal{S}_{\mathbf{z}} := \{\mathbf{z}^1, \ldots, \mathbf{z}^K\}$ with $\mathbf{z}^k \in \mathbb{R}^N$, the corresponding set of observables $\mathcal{S}_{\mathbf{x}} := \{\mathbf{x}^1, \ldots, \mathbf{x}^K\}$ is generated by $\mathcal{S}_{\mathbf{x}} = F(\mathcal{S}_{\mathbf{z}})$, where the map $F$ defines a push-forward measure $F_\#(P_{\mathcal{S}_{\mathbf{z}}})$ on the image of $F$ as: $F_\#(P_{\mathcal{S}_{\mathbf{z}}})(x_1, \ldots, x_K) = P_{\mathcal{S}_{\mathbf{z}}}(f_1^{-1}(x_1), \ldots, f_K^{-1}(x_K))$ with the support $\mathcal{X} := \text{Im}(F) \subseteq \mathbb{R}^{K \times D}$. Note that $F$ satisfies the diffeomorphism assumption (Asm. B.1) as each $f_k$ is a diffeomorphism onto its image according to Asm. B.1.

In the following, we denote by $\mathfrak{I} := \{\iota_i : \mathbb{R}^{|A_i|} \to \mathcal{M}_i\}$ a finite set of invariance properties with their respective invariant subsets $A_i \subseteq [N]$ and their equivalence relationships $\sim_{\iota_i}$, each inducing as a projection onto its quotient and invariant property $\iota_i$ (Defn. 2.1). For a set of observables $\mathcal{S}_{\mathbf{x}} := \{\mathbf{x}^1, \ldots, \mathbf{x}^K\} \in \mathcal{X}$ generated from the data generating process described in § 2, we assume:

**Assumption 2.1.** For each $\iota_i \in \mathfrak{I}$, there exists a ***unique known*** index subset $V_i \subseteq [K]$ with at least two elements (i.e., $|V_i| \geq 2$) s.t. $\mathbf{x}_{V_i} = F([\mathbf{z}]_{\sim_{\iota_i}})$ forms the set of observables generated from an equivalence class $[\mathbf{z}]_{\sim_{\iota_i}} := \{\tilde{\mathbf{z}} \in \mathbb{R}^N : \mathbf{z}_{A_i} \sim_{\iota_i} \tilde{\mathbf{z}}_{A_i}\}$, as given by Defn. 2.2. In particular, if $\mathfrak{I} = \{\iota\}$ consists of a single invariance property $\iota : \mathbb{R}^{|A|} \to \mathcal{M}$, we have $\mathcal{S}_{\mathbf{x}} = F([\mathbf{z}]_{\sim_\iota})$.

**Remark:** While $\mathfrak{I}$ does not need to be fully described, which observables should belong to the same equivalence class is known (denoted as $V_i \subseteq [K]$ for the invariance property $\iota_i \in \mathfrak{I}$). This

is a standard assumption and is equivalent to knowing e.g., two views are generated from partially overlapped latents [18].

Given a set of observables $\mathcal{S}_\mathbf{x} \in \mathcal{X}$ satisfying Asm. 2.1, we show that we can simultaneously identify multiple invariant latent blocks $A_i$ under a set of weak assumptions. In the best case, if each individual latent component is represented as a single invariant block through individual invariance property $\iota_i \in \mathfrak{I}$, we can learn a fully disentangled representation and further identify the latent causal graph by additional technical assumptions.

## 3 Identifiability Theory via the Invariance Principle

**High-level overview.** This section presents a general theory for latent variable identification that brings together many identifiability results from existing CRL works, including multiview, interventional, temporal, and multitask CRL. Our theory of latent variable identifiability, based on the invariance principle, consists of two key components: (1) ensuring the encoder's sufficiency, thereby obtaining an adequate representation of the original input for the desired task; (2) guaranteeing the learned representation to preserve known data symmetries as invariance properties. The sufficiency is often enforced by minimizing the reconstruction loss [8–10, 19, 20] in auto-encoder based architecture, maximizing the log likelihood in normalizing flows or maximizing entropy [18, 21–23] in contrastive-learning based approaches. The invariance property in the learned representations is often enforced by minimizing some equivalence relation-induced pseudometric between a pair of encodings [6, 10, 18, 22] or by some iterative algorithm that provably ensures the invariance property on the output [24, 25]. As a result, all invariant blocks $A_i, i \in [n_\mathfrak{I}]$ can be identified up to a mixing within the blocks while being disentangled from the rest. This type of identifiability is defined as *block-identifiability* [22] which we restate as follows:

**Definition 3.1** (Block-identifiability [22]). A subset of latent variable $\mathbf{z}_A := \{\mathbf{z}_j\}_{j \in A}$ with $A \subseteq [N]$ is block-identified by an encoder $g : \mathbb{R}^D \to \mathbb{R}^N$ on the invariant subset $A$ if the learned representation $\hat{\mathbf{z}}_{\hat{A}} := [g(\mathbf{x})]_{\hat{A}}$ with $\hat{A} \subseteq [N], |A| = |\hat{A}|$ contains all and only information about the ground truth $\mathbf{z}_A$, i.e. $\hat{\mathbf{z}}_{\hat{A}} = h(\mathbf{z}_A)$ for some diffeomorphism $h : \mathbb{R}^{|A|} \to \mathbb{R}^{|A|}$.

**Definition 3.2** (Encoders). The encoders $G := \{g_k : \mathcal{X}^k \to \mathcal{Z}^k\}_{k \in [K]}$ consist of smooth functions mapping from the observational support $\mathcal{X}^k$ to the corresponding latent support $\mathcal{Z}^k$ (§ 2).

**Definition 3.3** (Selection [18]). A selection $\oslash$ operates between two vectors $a \in \{0,1\}^d, b \in \mathbb{R}^d$ s.t. $a \oslash b := [b_j : a_j = 1, j \in [d]]$.

**Definition 3.4** (Invariant block selectors). The invariant block selectors $\Phi := \{\phi^{(i,k)}\}_{i \in [n_\mathfrak{I}], k \in V_i}$ with $\phi^{(i,k)} \in \{0,1\}^N$ perform selection (Defn. 3.3) on the encoded information: for any invariance property $\iota_i \in \mathfrak{I}$, any observable $\mathbf{x}^k, k \in V_i$ we have the selected representation:

$$\phi^{(i,k)} \oslash \hat{\mathbf{z}}^k = \phi^{(i,k)} \oslash g_k(\mathbf{x}^k) = \left[[g_k(\mathbf{x}^k)]_j : \phi_j^{(i,k)} = 1, j \in [N]\right], \tag{3.1}$$

with $\left\|\phi^{(i,k)}\right\|_0 = \left\|\phi^{(i,k')}\right\|_0 = |A_i|$ for all $\iota_i \in \mathfrak{I}, k, k' \in V_i$.

**Constraint 3.1** (Invariance constraint). *For any invariance property $\iota_i \in \mathfrak{I}, i \in [n_\mathfrak{I}]$, the **selected** representations $\phi^{(i,k)} \oslash g_k(\mathbf{x}^k), k \in V_i$ must be $\iota_i$-invariant across the observables from the subset $V_i \subseteq [K]$:*

$$\iota_i(\phi^{(i,k)} \oslash g_k(\mathbf{x}^k)) = \iota_i(\phi^{(i,k')} \oslash g_{k'}(\mathbf{x}^{k'})) \quad \forall i \in [n_\mathfrak{I}] \ \forall k, k' \in V_i \tag{3.2}$$

**Constraint 3.2** (Sufficiency constraint). *For any $\iota_i \in \mathfrak{I}, i \in [n_\mathfrak{I}]$, the **selected** representation $\phi^{(i,k)} \oslash g_k(\mathbf{x}^k), k \in V_i$ must preserve all information of the invariant partition $\mathbf{z}_{A_i}$ that we aim to identify, i.e., $I(\mathbf{z}_{A_i}, \phi^{(i,k)} \oslash g_k(\mathbf{x}^k)) = H(\mathbf{z}_{A_i}) \ \forall i \in [n_\mathfrak{I}], k \in V_i$.*

**Theorem 3.1** (Identifiability of multiple invariant blocks). *Consider a set of observables $\mathcal{S}_\mathbf{x} = \{\mathbf{x}^1, \mathbf{x}^2, \ldots, \mathbf{x}^K\}$ with $\mathbf{x}^k \in \mathcal{X}^k$ generated from § 2 satisfying Asm. 2.1. Let $G, \Phi$ be the set of smooth encoders (Defn. 3.2) and selectors (Defn. 3.4) that satisfy Constraints 3.1 and 3.2, then the invariant component $\mathbf{z}_{A_i}^k$ is block-identified (Defn. 3.1) by $\phi^{(i,k)} \oslash g_k$ for all $\iota_i \in \mathfrak{I}, k \in [K]$.*

**What about the variant latents?** Intuitively, the variant latents are generally not identifiable, as the invariance constraint (Constraint 3.1) is applied only to the selected invariant encodings, leaving the variant part without any weak supervision [26]. This result is formalized as follows:

**Proposition 3.2** (General non-identifiability of variant latent variables). *Consider the setup in Thm. 3.1, let $A := \bigcup_{i \in [n_{\mathfrak{I}}]} A_i$ denote the union of block-identified latent indices and $A^{\mathrm{c}} := [N] \setminus A$ the complementary set where no $\iota$-invariance $\iota \in \mathfrak{I}$ applies, then the variant latents $\mathbf{z}_{A^{\mathrm{c}}}$ cannot be identified.*

Although variant latent variables are generally non-identifiable, they can be identified under certain conditions. The following demonstrates that variant latent variables can be identified under invertible encoders when the variant and invariant partitions are mutually independent.

**Proposition 3.3** (Identifiability of variant latent under independence). *Consider an optimal encoder $g \in G^*$ and optimal selector $\phi \in \Phi^*$ from Thm. 3.1 that jointly identify an invariant block $\mathbf{z}_A$ (we omit subscriptions $k, i$ for simplicity), then $\mathbf{z}_{A^{\mathrm{c}}} (A^{\mathrm{c}} := [N] \setminus A)$ can be identified by the complementary encoding partition $(1 - \phi) \oslash g$ only if: (i) $g$ is invertible in the sense that $I(\mathbf{x}, g(\mathbf{x})) = H(\mathbf{x})$; (ii) $\mathbf{z}_{A^{\mathrm{c}}}$ is independent on $\mathbf{z}_A$.*

## 4 Experiments

This section demonstrates the real-world applicability of causal representation learning under the invariance principle, evidenced by superior treatment effect estimation performance on the high-dimensional causal inference benchmark [17] using a loss for the domain generalization literature that utilizes the invariance principle [13] (§ 4). Additionally, we provide ablation studies on existing interventional causal representation learning methods [2, 3, 27], showcasing that non-trivial distributional invariance is needed for latent variable identification. This distributional invariance could, but does not have to arise from a valid intervention in the sense of causality (App. F).

**Case Study: ISTAnt**  This experiment focuses on ISTAnt [17], a recent real-world ecological benchmark designed for treatment effect estimation. ISTAnt consists of video recordings of ants triplets with occasional grooming behavior. The goal is to extract a per-frame representation for supervised behavior classification (grooming or not) to estimate the Average Treatment Effect of an intervention (exposure to a chemical substance). Further details about this dataset and problem setting is provided in App. G.1

**Experiment settings.**  Different videos in ISTAnt are considered different *experiments* as the experiment settings and treatments vary. We consider hard annotation sampling criteria (more non-annotated than annotated) for both experiments (videos) and positions, as described by [17]. For the training, we adopt a domain generalization objective that utilizes the invariance principle [13], which is restated as follows:

$$\mathcal{R}_{\text{V-REx}}(\mathbf{w} \circ g) = \underbrace{\lambda_{\text{INV}} \operatorname{Var}(\{\mathcal{R}_1(\mathbf{w} \circ g), \dots, \mathcal{R}_K(\mathbf{w} \circ g)\})}_{\text{invariance}} + \underbrace{\sum_{k \in [K]} \mathcal{R}_k(\mathbf{w} \circ g)}_{\text{sufficiency}}. \tag{4.1}$$

We vary the strength of the invariant component in eq. (4.1) by setting the invariance regularization multiplier $\lambda_{\text{INV}}$ from 0 (ERM) to 10 000. We repeat 20 independent runs for each $\lambda_{\text{INV}}$ to estimate the statistical error. All other implementational details follow [17]. We evaluate the performance with both *balanced accuracy* and *Treatment Effect Relative Bias* (TERB). TERB is defined in [17] as the ratio between the bias in the predictions across treatment groups and the true average treatment effect estimated with ground-truth annotations over the whole trial.

**Results.**  Fig. 1 depicts the model performance regarding varying invariance regularization strength $\lambda_{\text{INV}}$. As expected, the balanced accuracy initially increases with the invariance regularization strength $\lambda_{\text{INV}}$, as our prediction problem benefits from the invariance, until the sufficiency component is not sufficiently balanced with the invariance, and performance decreases. Similarly, the TERB improves positively, weighting the invariance component until a certain threshold. In particular, on average with $\lambda_{\text{INV}} = 100$ the TERB decreases to 20% (from 100% using ERM) with experiment subsampling. In agreement with [17], a naive estimate of the TEB on a small validation set is a reasonable (albeit not perfect) model selection criterion. Although it performs slightly worse than model selection based on ERM loss in the position sampling case, it proves to be more reliable overall. This experiment underscores the advantages of flexibly enforcing known invariances in the data, corroborating our identifiability theory (§ 3).

## 5 Conclusions

In this paper, we take a closer look at the wide range of CRL methods. Interestingly, we find many CRL approaches share methodological similarities in aligning the representation to known data sym-

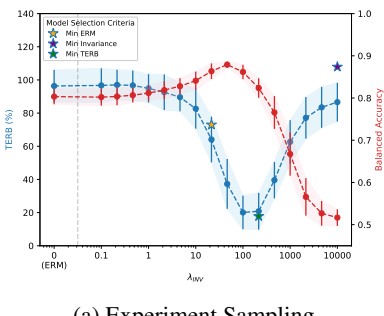 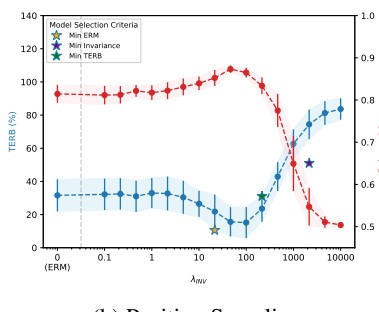

| (a) Experiment Sampling | (b) Position Sampling |

Figure 1: TERB and Balanced Accuracy with standard deviation over 20 different seeds varying the invariance weight $\lambda_{\text{INV}}$ of V-REx [13] on ISTAnt dataset [17]. With stars, the TERB of the model is selected by different model selection criteria on a small but heterogeneous validation set.

metries. We identified two components involved in identifiability results: preserving information of the data and a set of known invariances (§ 3). Our results help clarify the role of causal assumptions in causal variable identification, shifting the focus from a characterization of specific assumptions for identifiability, which are not necessarily satisfied in real-world scenarios, to a general recipe that allows practitioners to specify known invariances in their problem and learn representations that align with them. We successfully exemplified the real-world applicability of CRL on ecological data, as shown in § 4. Nevertheless, our paper leaves out certain settings concerning identifiability that may be interesting for future work, such as discrete variables and finite sample guarantees.

## Acknowledgements

We thank Jiaqi Zhang, Francesco Montagna, David Lopez-Paz, Kartik Ahuja, Thomas Kipf, Sara Magliacane, Julius von Kügelgen, Kun Zhang, and Bernhard Schölkopf for extremely helpful discussion. Riccardo Cadei was supported by a Google Research Scholar Award to Francesco Locatello. We acknowledge the Third Bellairs Workshop on Causal Representation Learning held at the Bellairs Research Institute, February 9/16, 2024, and a debate on the difference between interventions and counterfactuals in disentanglement and CRL that took place during Dhanya Sridhar's lecture, which motivated us to significantly broaden the scope of the paper. We thank Dhanya and all participants of the workshop.

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

# Appendix

## Table of Contents

## A  Notation and Terminology

$[N]$ is used as a shorthand for $\{1, \ldots, N\}$. We use bold lower-case $\mathbf{z}$ for random vectors and normal lower-case $z$ for their realizations. A vector $\mathbf{z}$ can be indexed either by a single index $i \in [\dim(\mathbf{z})]$ via $\mathbf{z}_i$ or a index subset $A \subseteq [\dim(\mathbf{z})]$ with $\mathbf{z}_A := \{\mathbf{z}_i : i \in A\}$. $P_{\mathbf{z}}$ denotes the probability distribution of the random vector $\mathbf{z}$ and $p_{\mathbf{z}}(z)$ denotes the associated probability density function. By default, a "measurable" function is *measurable* w.r.t. the Borel sigma algebras and defined w.r.t. the Lebesgue measure. A comprehensive list of notation follows:

| | |
|---|---|
| $f$ | Mixing function |
| $g$ | Smooth encoder |
| $\mathcal{G}$ | Ground truth causal graph |
| $\mathbf{x}$ | Entangled observables |
| $\mathbf{z}$ | Ground truth latent variables |
| $D$ | Dimensionality of observable $\mathbf{x}$ |

| | |
|---|---|
| $N$ | Dimensionality of latents $\mathbf{z}$ |
| $A$ | Subset of latent indices with invariance properties ($A \subseteq [N]$) |
| $\iota$ | Projector which maps the latents to the space where the invariance property holds |
| $\sim_\iota$ | The latent equivalence relation |
| $\mathfrak{I}$ | A set of invariance properties |
| $\mathcal{X}$ | Support of a set of observables $\mathcal{S}_{\mathbf{x}}$ |
| $\mathcal{Z}$ | Support of a set of latent vectors $\mathcal{S}_{\mathbf{z}}$ |
| $G$ | A set of smooth encoders |
| $\Phi$ | A set of selectors |
| TC | Transitive closure |

## B    Preliminaries

In this subsection, we revisit the common definitions and assumptions in identifiability works from causal representation learning. We begin with the definition of a latent structural causal model:

**Definition B.1** (Latent SCM [2]). Let $\mathbf{z} = \{\mathbf{z}_1, \ldots, \mathbf{z}_N\}$ denote a set of causal "endogenous" variables with each $\mathbf{z}_i$ taking values in $\mathbb{R}$, and let $\mathbf{u} = \{\mathbf{u}_1, \ldots, \mathbf{u}_N\}$ denotes a set of mutually independent "exogenous" random variables. The latent SCM consists of a set of structural equations

$$\{\mathbf{z}_i := m_i(\mathbf{z}_{\mathrm{pa}(i)}), \mathbf{u}_i\}_{i=1}^N, \tag{B.1}$$

where $\mathbf{z}_{\mathrm{pa}(i)}$ are the causal parents of $\mathbf{z}_i$ and $m_i$ are the deterministic functions that are termed "causal mechanisms". We indicate with $P_{\mathbf{u}}$ the joint distribution of the exogenous random variables, which due to the independence hypothesis is the product of the probability measures of the individual variables. The associated causal diagram $\mathcal{G}$ is a directed graph with vertices $\mathbf{z}$ and edges $\mathbf{z}_i \to \mathbf{z}_j$ iff. $\mathbf{z}_i \in \mathbf{z}_{\mathrm{pa}(j)}$; we assume the graph $\mathcal{G}$ to be acyclic.

The latent SCM induces a unique distribution $P_{\mathbf{z}}$ over the endogenous variables $\mathbf{z}$ as a pushforward of $P_{\mathbf{u}}$ via eq. (B.1). Its density $p_{\mathbf{z}}$ follows the causal Markov factorization:

$$p_{\mathbf{z}}(z) = \prod_{i=1}^N p_i(z_i \mid z_{\mathrm{pa}(i)}). \tag{B.2}$$

Instead of directly observing the endogenous and exogenous variables $\mathbf{z}$ and $\mathbf{u}$, we only have access to some "entangled" measurements $\mathbf{x}$ of $\mathbf{z}$ generated through a nonlinear mixing function:

**Definition B.2** (Mixing function). A deterministic smooth function $f : \mathbb{R}^N \to \mathbb{R}^D$ mapping the latent vector $\mathbf{z} \in \mathbb{R}^N$ to its observable $\mathbf{x} \in \mathbb{R}^D$, where $D \geq N$ denotes the dimensionality of the observational space.

**Assumption B.1** (Diffeomorphism). The mixing function $f$ is diffeomorphic onto its image, i.e. $f$ is $C^\infty$, $f$ is injective and $f^{-1}|_{\mathcal{I}(f)} : \mathcal{I}(f) \to \mathbb{R}^D$ is also $C^\infty$.

**Remark:** Settings with noisy observations ($\mathbf{x} = f(\mathbf{z}) + \epsilon$, $\mathbf{z} \perp \epsilon$) can be easily reduced to our denoised version by applying a standard deconvolution argument as a pre-processing step, as indicated by Buchholz et al. [5], Lachapelle et al. [8].

## C    Identifiability Theory

In addition to the general results for latent variable identification presented in § 3, we compare in App. C.1 different granularity of latent variable identification and show their transitions through certain assumptions on the causal model or mixing function. Afterward, App. C.2 discusses the identification level of a causal graph depending on the granularity of latent variable identification under certain structural assumptions. Detailed proofs are deferred to App. E.

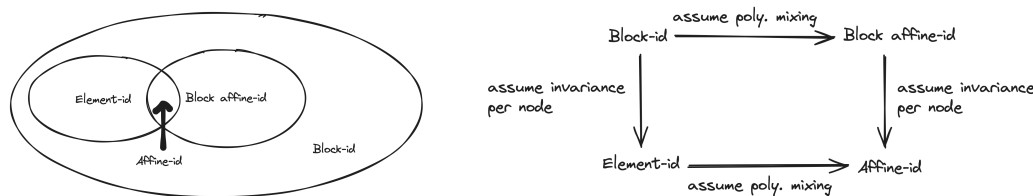

Figure 2: Relations between different identification classes (Defns. 3.1 and C.1 to C.3). Some CRL works proposed a more fine-grained classification of identifiability concepts with slightly different terminology, which we omit here for readability.

## C.1 On the granularity of identification

Different levels of identification can be achieved depending on the degree of underlying data symmetry. Below, we present three standard identifiability definitions from the CRL literature, each offering stronger identification results than block-identifiability (Defn. 3.1).

**Definition C.1** (Block affine-identifiability). Let $\hat{\mathbf{z}}$ be the learned representation, for a subset $A \subseteq [N]$ it satisfies that:

$$\hat{\mathbf{z}}_{\pi(A)} = D \cdot \mathbf{z}_A + \mathbf{b}, \tag{C.1}$$

where $D \in \mathbb{R}^{|A| \times |A|}$ is an invertible matrix, $\pi(A)$ denotes the index permutation of $A$, then $\mathbf{z}_A$ is block affine-identified by $\hat{\mathbf{z}}_{\pi(A)}$.

**Definition C.2** (Element-identifiability). The learned representation $\hat{\mathbf{z}} \in \mathbb{R}^N$ satisfies that:

$$\hat{\mathbf{z}} = \mathbf{P}_\pi \cdot h(\mathbf{z}), \tag{C.2}$$

where $\mathbf{P}_\pi \in \mathbb{R}^{N \times N}$ is a permutation matrix, $h(\mathbf{z}) := (h_1(\mathbf{z}_1), \ldots h_N(\mathbf{z}_N)) \in \mathbb{R}^N$ is a is an element-wise diffeomorphism.

**Definition C.3** (Affine-identifiability). The learned representation $\hat{\mathbf{z}} \in \mathbb{R}^N$ satisfies that:

$$\hat{\mathbf{z}} = \Lambda \cdot \mathbf{P}_\pi \cdot \mathbf{z} + \mathbf{b}, \tag{C.3}$$

where $\mathbf{P}_\pi \in \mathbb{R}^{N \times N}$ is a permutation matrix, $\Lambda \in \mathbb{R}^{N \times N}$ is a diagonal matrix with nonzero diagonal entries.

> **Remark**: Block affine-identifiability (Defn. C.1) is defined by Ahuja et al. [3], stating that the learned representation $\hat{\mathbf{z}}$ is related to the ground truth latents $\mathbf{z}$ through some sparse matrix with zero blocks. Defn. C.2 indicates element-wise identification of latent variables up to individual diffeomorphisms. Element-identifiability for the latent variable identification together with the graph identifiability (Defn. C.4) is defined as $\sim_{\text{CRL}}$-identifiability [2, Defn. 2.6], perfect identifiability [7, Defn. 3]. Affine identifiability (Defn. C.3) describes when the ground truth latent variables are identified up to permutation, shift, and linear scaling. In many CRL works, affine identifiability (Defn. C.3) is also termed as follows: perfect identifiability under linear transformation [25, Defn. 1], CD-equivalence [6, Defn. 1], disentanglement [8, Defn. 3].

**Proposition C.1** (Granularity of identification). *Affine-identifiability (Defn. C.3) implies element-identifiability (Defn. C.2) and block affine-identifiability (Defn. C.1) while element-identifiability and block affine-identifiability implies block-identifiability (Defn. 3.1).*

**Proposition C.2** (Transition between identification levels). *The transition between different levels of latent variable identification (Fig. 2) can be summarized as follows:*

*(i) Element-level identifiability (Defns. C.2 and C.3) can be obtained from block-wise identifiability (Defns. 3.1 and C.1) when each individual latent constitutes an invariant block;*

*(ii) Identifiability up to an affine transformation (Defns. C.1 and C.3) can be obtained from general identifiability on arbitrary diffeomorphism (Defns. 3.1 and C.2) by additionally assuming that both the ground truth mixing function and decoder are finite degree polynomials of the same degree.*

**Discussion.** We note that the granularity of identifiability results is primarily determined by the strength of invariance and parametric assumptions (such as those on mixing functions or causal models) rather than by the specific algorithmic choice. For example, for settings that can achieve element-identifiability [2], affine-identifiability results can be obtained by additionally assuming *finite degree polynomial* mixing function (proof see App. E). Similarly, one reaches element-identifiability from block-identifiability by enforcing invariance properties on each latent component [18, Thm. 3.8] instead of having only *one* fat-hand invariant block [22]. Tab. 2 provides an overview of recent identifiability results along with their corresponding invariance and parametric assumptions, illustrating the direct relationship between these assumptions and the level of identifiability they achieve.

### C.2  Identifying the causal graph

In addition to latent variable identification, another goal of causal representation learning is to infer the underlying latent dependency, namely the causal graph structure. Hence, we restate the standard definition of graph identifiability in causal representation learning.

**Definition C.4** (Graph-identfiability). The estimated graph $\hat{\mathcal{G}}$ is isomorphic to the ground truth $\mathcal{G}$ through a bijection $h : V(\mathcal{G}) \to V(\hat{\mathcal{G}})$ in the sense that two vertices $\mathbf{z}_i, \mathbf{z}_j \in V(\mathcal{G})$ are adjacent in $\mathcal{G}$ if and only if $h(\mathbf{z}_i), h(\mathbf{z}_j) \in V(\hat{\mathcal{G}})$ are adjacent in $\hat{\mathcal{G}}$.

We remark that the "faithfulness" assumption [28, Defn. 2.4.1] is a standard assumption in the CRL literature, commonly required for graph discovery. We restate it as follows:

**Assumption C.1** (Faithfulness (or Stability)). $P_{\mathbf{z}}$ is a faithful distribution induced by the latent SCM (Defn. B.1) in the sense that $P_{\mathbf{z}}$ contains no extraneous conditional independence; in other words, the only conditional independence relations satisfied by $P_{\mathbf{z}}$ are those given by $\{\mathbf{z}_i \perp \mathbf{z}_{\text{nd}(i)} \mid \mathbf{z}_{\text{pa}(i)}\}$ where $\mathbf{z}_{\text{nd}(i)}$ denotes the non-descends of $\mathbf{z}_i$.

As indicated by Defn. C.4, the preliminary condition of identifying the causal graph is to have an element-wise correspondence between the vertices in the ground truth graph $\mathcal{G}$ (i.e., the ground truth latents) and the vertices of the estimated graph. Therefore, the following assumes that the learned encoders $G$ (Defn. 3.2) achieve element-identifiability (Defn. C.2), that is, for each $\mathbf{z}_i \in \mathbf{z}$, we have a diffeomorphism $h_i : \mathbb{R} \to \mathbb{R}$ such that $\hat{\mathbf{z}}_i = h_i(\mathbf{z}_i)$. However, to identify the graph structure, additional assumptions are needed: either on the source of invariance or on the parametric form of the latent causal model.

**Graph identification via interventions.** Under the element-identifiability (Defn. C.2) of the latent variables $\mathbf{z}$, the causal graph structure $\mathcal{G}$ can be identified up to its isomorphism (Defn. C.4), given multi-domain data from *paired perfect* interventions per-node [2, 7]. Using data generated from *imperfect* interventions is generally insufficient to identify the direct edges in the causal graph, it can only identify the ancestral relations, i.e., up to the transitive closure of $\mathcal{G}$ [4, 6]. Unfortunately, even imposing the linear assumption on the latent SCM does not provide a solution [24]. Nevertheless, by adding sparsity assumptions on the causal graph $\mathcal{G}$ and polynomial assumption on the mixing function $f$, Zhang et al. [6] has shown isomorphic graph identifiability (Defn. C.4) under *imperfect* intervention per node. In general, access to the interventions is necessary for graph identification if one is not comfortable making other parametric assumptions about the graph structure. Conveniently, in this setting, the graph identifiability is linked with that of the variables since the latter leverages the invariance induced by the intervention.

**Graph identification via parametric assumptions.** It is well known in causal discovery that the additive noise model [29] is identifiable under certain mild assumptions [30, 31]. In the following, we assume an additive exogenous noise in the latent SCM (Defn. B.1):

**Assumption C.2** (Additive noise). The endogenous variable $\mathbf{z}_i \in \mathbb{R}$ in the previously defined latent SCM (Defn. B.1) relates to the corresponding exogenous noise variable $\mathbf{u}_i \in \mathbb{R}$ through additivity. Namely, the causal mechanism (eq. (B.1)) can be rewritten as:

$$\{\mathbf{z}_i = m_i(\mathbf{z}_{\text{pa}(i)}) + \mathbf{u}_i\}. \tag{C.4}$$

As a generalization of the additive noise model, the post-nonlinear acyclic causal model [30, Sec. 2] allows extra nonlinearity on the top of the additive causal mechanism, providing additional flexibility on the latent model assumption:

**Definition C.5** (Post-nonlinear acyclic causal model). The following causal mechanism describes a post-nonlinear acyclic causal model:

$$\mathbf{z}_i = h_i(m_i(\mathbf{z}_{\mathrm{pa}(i)}) + \mathbf{u}_i), \tag{C.5}$$

where $h_i : \mathbb{R} \to \mathbb{R}$ is a diffeomorphism and $m_i$ is a non-constant function.

Assume the latent variable $\mathbf{z}_i$ is element-wise identified through a bijective mapping $h_i : \mathbb{R} \to \mathbb{R}$ for all $i \in [N]$, define the estimated causal parents $\hat{\mathbf{z}}_{\mathrm{pa}(i)} := \{h_j(\mathbf{z}_j) : \mathbf{z}_j \in \mathbf{z}_{\mathrm{pa}(i)}\}$, then the latent SCM (Defn. B.1) is translated to a post-nonlinear acyclic causal model (Defn. C.5) because

$$
\begin{aligned}
\hat{\mathbf{z}}_i = h_i(\mathbf{z}_i) &= h_i(m_i(\mathbf{z}_{\mathrm{pa}(i)}) + \mathbf{u}_i) \\
&= h_i(m_i(\{h_j^{-1}(\hat{\mathbf{z}}_j) : \mathbf{z}_j \in \mathbf{z}_{\mathrm{pa}(i)}\}) + \mathbf{u}_i) \\
&= h_i(\tilde{m}_i(\hat{\mathbf{z}}_{\mathrm{pa}(i)}) + \mathbf{u}_i),
\end{aligned}
\tag{C.6}
$$

where

$$\tilde{m}_i(\hat{\mathbf{z}}_{\mathrm{pa}(i)}) := m_i(\{h_j^{-1}(\hat{\mathbf{z}}_j) : \mathbf{z}_j \in \mathbf{z}_{\mathrm{pa}(i)}\}).$$

Thus, the underlying causal graph $\mathcal{G}$ can be identified up to an isomorphism (Defn. C.4) following the approach given by Zhang and Hyvärinen [31, Sec. 4]

**What happens if variables are identified in blocks?** Consider the case where the latent variables cannot be identified up to element-wise diffeomorphism; instead, one can only obtain a coarse-grained version of the variables (e.g., as a mixing of a block of variables (Defn. 3.1)). Nevertheless, certain causal links between these coarse-grained block variables are of interest. These block variables and their causal relations in between form a "macro" level of the original latent SCM, which is shown to be causally consistent under mild structural assumptions [32, Thm. 11]. In particular, the macro-level model can be obtained from the micro-level model through an *exact transformation* [33, Defn. 3.4] and thus produces the same causal effect as the original micro-level model under the same type of interventions, providing useful knowledge for downstream causal analysis. More formal connections are beyond the scope of this paper. Still, we see this concept of coarse-grained identification on both causal variables and graphs as an interesting avenue for future research.

## D   Related Works

This section reviews related causal representation learning works and frames them as specific instances of our theory (§ 3). These works were originally categorized into various causal representation learning types (multiview, multi-domain, multi-task, and temporal CRL) based on the level of invariance in the data-generating process, leading to varying degrees of identifiability results (App. C.1). While the implementation of individual works may vary, the *methodological principle of aligning representation with known data symmetries* remain consistent, as shown in § 3. We begin with revisiting the data-generating process of each category and explain how they can be viewed as specific cases of the proposed invariance framework (§ 2). We then present individual identification algorithms from the CRL literature as particular applications of our theorems based on the implementation choices needed to satisfy the invariance and sufficiency constraints (Constraints 3.1 and 3.2). A more detailed overview of the individual works is provided in Tab. 2.

### D.1   Multiview Causal Representation Learning

**High-level overview.** The multiview setting in causal representation learning [18, 23] considers multiple views that are *concurrently* generated by an overlapping subset of latent variables, and thus having *non-independently* distributed data. Multiview scenarios are often found in a partially observable setup. For example, multiple devices on a robot measure different modalities, jointly monitoring the environment through these real-time measurements. While each device measures a distinct subset of latent variables, these subsets probably still overlap as they are measuring the same system at the same time. In addition to partial observability, another way to obtain multiple views is to perform an "intervention/perturbation" [4, 19, 20, 22] and collect both pre-action and post-action views on the same sample. This setting is often improperly termed "counterfactual"[1] in the CRL

---

[1]Traditionally, counterfactual in causality refers to non-observable outcomes that are "counter to the fact" [34]. In the works we refer to here, they rather represent pre- and post- an action that affect some latent variables but not all. This can be mathematically expressed as a counterfactual in a SCM, but is conceptually different as both pre- and post- action outcomes are realized [35]. The "counterfactual" terminology silently implies that this is a strong assumption, but nuance is needed and it can in fact be much weaker than an intervention.

literature, and this type of data is termed "paired data". From another perspective, the paired setting can be cast in the partial observability scenario by considering the same latent before and after an action (mathematically modelled as an intervention) as two separate latent nodes in the causal graph, as shown by von Kügelgen et al. [22, Fig. 1]. Thus, both pre-action and post-action views are partial because neither of them can observe pre-action and post-action latents simultaneously. These works assume that the latents that are not affected by the action remain constant, an assumption that is relaxed in temporal CRL works. See App. D.3 for more discussion in this regard.

**Data generating process.** In the following, we introduce the data-generating process of a multi-view setting in the flavor of the invariance principle as introduced in § 2. We consider a set of views $\{\mathbf{x}^k\}_{k \in [K]}$ with each view $\mathbf{x}^k \in \mathcal{X}^k$ generated from some latents $\mathbf{z}^k \in \mathcal{Z}^k$. Let $S_k \subseteq [N]$ be the index set of generating factors for the view $\mathbf{x}^k$, we define $\mathbf{z}_j^k = 0$ for all $j \in [N] \setminus S_k$ to represent the uninvolved partition of latents. Each entangled view $\mathbf{x}^k$ is generated by a view-specific mixing function $f_k : \mathcal{Z}^k \to \mathcal{X}^k$:

$$\mathbf{x}^k = f_k(\mathbf{z}^k) \quad \forall k \in [K] \tag{D.1}$$

Define the joint overlapping index set $A := \bigcap_{k \in [K]} S_k$, and assume $A \subseteq [N]$ is a non-empty interior of $[N]$. Then the value of the sharing partition $\mathbf{z}_A$ remain ***invariant*** for all observables $\{\mathbf{x}^k\}_{k \in [K]}$ on a ***sample level***. By considering the joint intersection $A$, we have *one single* invariance property $\iota : \mathbb{R}^{|A|} \to \mathbb{R}^{|A|}$ in the invariance set $\mathfrak{I}$; and this invariance property $\iota$ emerges as the identity map id on $\mathbb{R}^{|A|}$ in the sense that $\mathrm{id}(\mathbf{z}_A^k) = \mathrm{id}(\mathbf{z}_A^{k'})$ and thus $\mathbf{z}_A^k \sim_\iota \mathbf{z}_A^{k'}$ for all $k, k' \in [K]$. Note that Defn. 2.1 (ii) is satisfied because any transformation $h_k$ that involves other components $\mathbf{z}_q$ with $q \notin A$ violates the equity introduced by the identity map. For a subset of observations $V_i \subseteq [K]$ with at least two elements $|V_i| > 1$, we define the latent intersection as $A_i := \bigcap_{k \in V_i} \subseteq [N]$, then for each non-empty intersection $A_i$, there is a corresponding invariance property $\iota_i : \mathbb{R}^{|A_i|} \to \mathbb{R}^{|A_i|}$ which is the identity map specified on the subspace $\mathbb{R}^{|A_i|}$. By considering all these subsets $\mathcal{V} := \{V_i \subseteq [K] : |V_i| > 1, |A_i| > 0\}$, we obtain a set of invariance properties $\mathfrak{I} := \{\iota_i : \mathbb{R}^{|A_i|} \to \mathbb{R}^{|A_i|}\}$ that satisfy Asm. 2.1.

**Identification algorithms.** Many multiview works [18, 22, 23] employ the $L_2$ loss as a regularizer to enforce ***sample-level*** invariance on the invariant partition, cooperated with some sufficiency regularizer to preserve sufficient information about the observables (Constraint 3.2). Aligned with our theory (Thm. 3.1), these works have shown block-identifiability on the invariant partition of the latents across different views. Following the same principle, there are certain variations in the implementations to enforce the invariance principle, e.g. Locatello et al. [19] directly average the learned representations from paired data $g(\mathbf{x}^1), g(\mathbf{x}^2)$ on the shared coordinates before forwarding them to the decoder; Ahuja et al. [20] enforces $L_2$ alignment up to a learnable sparse perturbation $\delta$. As each latent component constitutes a single invariant block in the training data, these two works *element-identifies* (Defn. C.2) the latent variables, as explained by Proposition C.2.

## D.2 Multi-environment Causal Representation Learning

**High-level overview.** Multi-environment / interventional CRL considers data generated from multiple environments with respective environment-specific data distributions; hence, the considered data is *independently* but *non-identically distributed*. In the scope of causal representation learning, multi-environment data is often instantiated through interventions on the latent structured causal model [5–7, 22, 24, 25, 36]. Recently, several papers attempt to provide a more general identifiability statement where multi-environment data is not necessarily originated from interventions; instead, they can be individual data distributions that preserve certain symmetries, such as marginal invariance or support invariance [37] or sufficient statistical variability [38].

**Data generating process** The following presents the data generating process described in most interventional causal representation learning works. Formally, we consider a set of *non-identically* distributed data $\{P_{\mathbf{x}^k}\}_{k \in [K]}$ that are collected from multiple environments (indexed by $k \in [K]$) with a shared mixing function $f : \mathbf{x}^k = f(\mathbf{z}^k)$ (Defn. B.2) satisfying Asm. B.1 and a shared latent SCM (Defn. B.1). Let $k = 0$ denote the non-intervened environment and $\mathcal{I}_k \subseteq [N]$ denotes the set of intervened nodes in $k$-th environment, the latent distribution $P_{\mathbf{z}^k}$ is associated with the density

$$p_{\mathbf{z}^k}(z^k) = \prod_{j \in \mathcal{I}_k} \tilde{p}(z_j^k \mid z_{\mathrm{pa}(j)}^k) \prod_{j \in [N] \setminus \mathcal{I}_k} p(z_j^k \mid z_{\mathrm{pa}(j)}^k), \tag{D.2}$$

where we denote by $p$ the original density and by $\tilde{p}$ the intervened density. Interventions naturally introduce various distributional invariance that can be utilized for latent variable identification: Under the intervention $\mathcal{I}_k$ in the $k$-th environment, we observe that both (1) the marginal distribution of $\mathbf{z}_A$ with $A := [N] \setminus \mathrm{TC}(\mathcal{I}_k)$, with TC denoting the transitive closure and (2) the score $[S(\mathbf{z}^k)]_{A'} := \nabla_{\mathbf{z}^k_{A'}} \log p_{\mathbf{z}^k}$ on the subset of latent components $A' := [N] \setminus \overline{\mathrm{pa}}(\mathcal{I}_k)$ with $\overline{\mathrm{pa}}(\mathcal{I}_k) := \{j : j \in \mathcal{I}_k \cup \mathrm{pa}(\mathcal{I}_k)\}$ remain *invariant* across the observational and the $k$-th interventional environment. Formally, under intervention $\mathcal{I}_k$, we have

- *Marginal invariance*:

$$p_{\mathbf{z}^0}(z_A^0) = p_{\mathbf{z}^k}(z_A^k) \qquad A := [N] \setminus \mathrm{TC}(\mathcal{I}_k); \tag{D.3}$$

- *Score invariance*:

$$[S(\mathbf{z}^0)]_{A'} = [S(\mathbf{z}^k)]_{A'} \qquad A' := [N] \setminus \overline{\mathrm{pa}}(\mathcal{I}_k). \tag{D.4}$$

According to our theory Thm. 3.1, we can block-identify both $\mathbf{z}_A, \mathbf{z}'_A$ using these invariance principles (eqs. (D.3) and (D.4)). Since most interventional CRL works assume at least one intervention per node [2, 3, 5–7, 24, 36], more fine-grained variable identification results, such as element-wise identification (Defn. C.2) or affine-identification (Defn. C.3), can be achieved by combining multiple invariances from these per-node interventions, as we elaborate below.

**Identifiability with one intervention per node.** By applying Thm. 3.1, we demonstrate that latent causal variables $\mathbf{z}$ can be identified up to element-wise diffeomorphism (Defn. C.2) under single node *imperfect* intervention per node, given the following assumption.

**Assumption D.1** (Topologically ordered interventional targets). Specifying Asm. 2.1 in the interventional setting, we assume there are exactly $N$ environments $\{k_1, \ldots, k_N\} \subseteq [K]$ where each node $j \in [N]$ undergoes one imperfect intervention in the environment $k_j \in [K]$. The interventional targets $1 \preceq \cdots \preceq N$ preserve the topological order, meaning that $i \preceq j$ only if there is a directed path from node $i$ to node $j$ in the underlying causal graph $\mathcal{G}$.

**Remark:** Asm. D.1 is directly implied by Asm. 2.1 as we need to know which environments fall into the same equivalence class. We believe that identifying the topological order is another subproblem orthogonal to identifying the latent variables, which is often termed "uncoupled/non-aligned problem" [2, 7]. As described by Zhang et al. [6], the topological order of unknown interventional targets can be recovered from single-node imperfect intervention by iteratively identifying the interventions that target the source nodes. This iterative identification process may require additional assumptions on the mixing functions [3, 6, 24, 25, 36] and the latent structured causal model [5, 24], or on the interventions, such as *perfect* interventions that eliminate parental dependency [7], or the need for two interventions per node [2, 7].

**Corollary D.1.** *Given $N$ environments $\{k_1, \ldots, k_N\} \subseteq [K]$ satisfying Asm. D.1, the ground truth latent variables $\mathbf{z}$ can be identified up to element-wise diffeomorphism (Defn. C.2) by combining both marginal and score invariances (eqs. (D.3) and (D.4)) under our framework (Thm. 3.1).*

*Proof.* We consider a coarse-grained version of the underlying causal graph consisting of a block-node $\mathbf{z}_{[N-1]}$ and the leaf node $\mathbf{z}_N$ with $\mathbf{z}_{[N-1]}$ causing $\mathbf{z}_N$ (i.e., $\mathbf{z}_{[N-1]} \to \mathbf{z}_N$). We first select a pair of environments $V = \{0, k_N\}$ consisting of the observational environment and the environment where the leaf node $\mathbf{z}_N$ is intervened upon. According to eq. (D.3), the *marginal invariance* holds for the partition $A = [N-1]$, implying identification on $\mathbf{z}_{[N-1]}$ from Thm. 3.1. At the same time, when considering the set of environments $V' = \{0, k_1, \ldots, k_{N-1}\}$, the leaf node $N$ is the only component that satisfy *score* invariance across all environments $V'$, because $N$ is not the parent of any intervened node (also see [36, Lemma 4]). So here we have another invariant partition $A' = \{N\}$, implying identification on $\mathbf{z}_N$ (Thm. 3.1). By jointly enforcing the marginal and score invariance on $A$ and $A'$ under a sufficient encoder (Constraint 3.2), we identify both $\mathbf{z}_{[N-1]}$ as a block and $\mathbf{z}_N$ as a single element. Formally, for the parental block $\mathbf{z}_{[N-1]}$, we have:

$$\hat{\mathbf{z}}_{[N-1]}^k = g_{:N-1}(\mathbf{x}^k) \qquad \forall k \in \{0, k_1, \ldots, k_N\} \tag{D.5}$$

where $g_{:N-1}(\mathbf{x}^k) := [g(\mathbf{x}^k)]_{:N-1}$ relates to the ground truth $\mathbf{z}_{[N-1]}$ through some diffeomorphism $h_{[N-1]} : \mathbb{R}^{N-1} \to \mathbb{R}^{N-1}$ (Defn. 3.1). Now, we can remove the leaf node $N$ as follows: For each

environment $k \in \{0, k_1, \ldots, k_{N-1}\}$, we compute the pushforward of $P_{\mathbf{x}^k}$ using the learned encoder $g_{:N-1} : \mathcal{X}^k \to \mathbb{R}^{N-1}$:

$$P_{\hat{\mathbf{z}}^k_{[N-1]}} = g_{\#}(P_{\mathbf{x}^k})$$

Note that the estimated representations $P_{\hat{\mathbf{z}}^k_{[N-1]}}$ can be seen as a new observed data distribution for each environment $k$ that is generated from the subgraph $\mathcal{G}_{-N}$ without the leaf node $N$. Using an iterative argument, we can identify all latent variables element-wise (Defn. C.2), concluding the proof. $\qquad\square$

Upon element-wise identification from single-node intervention per node, existing works often provide more fine-grained identifiability results by incorporating other parametric assumptions, either on the mixing functions [3, 6, 36] or the latent causal model [5] or both [24]. This is explained by Proposition C.2, as element-wise identification can be refined to affine-identification (Defn. C.3) given additional parametric assumptions. Note that under this milder setting, the full graph is not identifiable without further assumptions, see [6].

**Identifiability with two interventions per-node** Current literature in interventional CRL targeting the general nonparametric setting [2, 7] typically assumed a pair of *sufficiently different* perfect interventions per node. Thus, any latent variable $\mathbf{z}_j, j \in [N]$, as an interventional target, is ***uniquely shared*** by a pair of interventional environment $k, k' \in [K]$, forming an invariant partition $A_i = \{j\}$ constituting of individual latent node $j \in [N]$. Note that this invariance property on the interventional target induces the following distributional property:

$$[S(\mathbf{z}^k) - S(\mathbf{z}^{k'})]_j \neq 0 \qquad \text{only if} \qquad \mathcal{I}_k = \mathcal{I}_{k'} = \{j\}. \tag{D.6}$$

According to Thm. 3.1, each latent variable can thus be identified separately, giving rise to element-wise identification, as shown by [2, 7].

**Identifiability under multiple distributions.** More recently, Ahuja et al. [37] explains previous interventional identifiability results from a general weak distributional invariance perspective. In a nutshell, a set of variables $\mathbf{z}_A$ can be block-identified if certain invariant distributional properties hold: The invariant partition $\mathbf{z}_A$ can be block-identified (Defn. 3.1) from the rest by utilizing the *marginal distributional invariance* or *invariance on the support, mean or variance*. Ahuja et al. [37] additionally assume the mixing function to be finite degree polynomial, which leads to block-affine identification (Defn. C.1), whereas we can also consider a general non-parametric setting; they consider *one* single invariance set, which is a special case of Thm. 3.1 with one joint $\iota$-property.

**Identification algorithm.** Instead of iteratively enforcing the invariance constraint across the majority of environments as described in Cor. D.1, most single-node interventional works develop equivalent constraints between pairs of environments to optimize. For example, the marginal invariance (eq. (D.3)) implies the marginal of the source node is changed *only if* it is intervened upon, which is utilized by Zhang et al. [6] to identify latent variables and the ancestral relations simultaneously. In practice, Zhang et al. [6] propose a regularized loss that includes Maximum Mean Discrepancy(MMD) between the reconstructed "counterfactual" data distribution and the interventional distribution, enforcing the distributional discrepancy that reveals graphical structure (e.g., detecting the source node). Similarly, by enforcing sparsity on the score change matrix, Varici et al. [36] restricts only score changes from the intervened node and its parents. In the nonparametric case, von Kügelgen et al. [2] optimize for the invariant (aligned) interventional targets through model selection, whereas Varici et al. [7] directly solve the constrained optimization problem formulated using score differences. Considering a more general setup, Ahuja et al. [37] provides various invariance-based regularizers as plug-and-play components for any losses that enforce a sufficient representation (Constraint 3.2).

### D.3  Temporal Causal Representation Learning

**High-level overview.** Temporal CRL [8–11, 39–43] focuses on retrieving latent causal structures from time series data, where the latent causal structured is typically modeled as a Dynamic Bayesian Network (DBN) [44, 45]. Existing temporal CRL literature has developed identifiability results under varying sets of assumptions. A common overarching assumption is to require the Dynamic Bayesian Network to be first-order Markovian, allowing only causal links from $t-1$ to $t$, eliminating longer dependencies [9–11, 40]. While many works assume that there is no instantaneous effect, restricting the latent components of $\mathbf{z}^t$ to be mutually dependent [10, 11, 40], some approaches have

lifted this assumption and prove identifiability allowing for instantaneous links among the latent components at the same timestep (Lippe et al. [9]).

**Data generating process.** We present the data generating process followed by most temporal causal representation works and explain the underlying latent invariance and data symmetries. Let $\mathbf{z}^t \in \mathbb{R}^N$ denotes the latent vector at time $t$ and $\mathbf{x}^t = f(\mathbf{z}^t) \in \mathbb{R}^D$ the corresponding entangled observable with $f : \mathbb{R}^N \to \mathbb{R}^D$ the shared mixing function (Defn. B.2) satisfying Asm. B.1. The actions $\mathbf{a}^t$ with cardinality $|\mathbf{a}^t| = N$ mostly only target a subset of latent variables while keeping the rest untouched, following its default dynamics [8, 10, 11, 41]. Intuitively, these actions $\mathbf{a}^t$ can be interpreted as a component-wise indicator for each latent variable $\mathbf{z}_j^t, j \in [N]$ stating whether $\mathbf{z}_j$ follows the default dynamics $p(\mathbf{z}_j^{t+1} \mid \mathbf{z}^t)$ or the modified dynamics induced by the action $\mathbf{a}_j^t$. From this perspective, the non-intervened causal variables at time $t$ can be considered the invariant partition under our formulation, denoted by $\mathbf{z}_{A_t}^t$ with the index set $A_t$ defined as $A_t := \{j : \mathbf{a}_j = 0\}$. Note that this invariance can be considered as a generalization of the multiview case because the realizations $z_j^t, z_j^{t+1}$ are not exactly identical (as in the multiview case) but are related via a default transition mechanism $p(\mathbf{z}_j^{t+1} \mid \mathbf{z}^t)$. To formalize this intuition, we define $\tilde{\mathbf{z}}^t := \mathbf{z}^t \mid \mathbf{a}^t$ as the conditional random vector conditioning on the action $\mathbf{a}^t$ at time $t$. For the non-intervened partition $A_t \subseteq [N]$ that follows the default dynamics, the transition model should be invariant:

$$p(\mathbf{z}_{A_t}^t \mid \mathbf{z}^{t-1}) = p(\tilde{\mathbf{z}}_{A_t}^t \mid \mathbf{z}^{t-1}), \tag{D.7}$$

which gives rise to a non-trivial distributional invariance property (Defn. 2.1). Note that the invariance partition $A_t$ could vary across different time steps, providing a set of invariance properties $\mathfrak{I} := \{\iota_t : \mathbb{R}^{|A_t|} \to \mathcal{M}_t\}_{t=1}^T$, indexed by time $t$. Given by Thm. 3.1, all invariant partitions $\mathbf{z}_{A_t}^t$ can be block-identified; furthermore, the complementary variant partition can also be identified under an invertible encoder and mutual independence within $\mathbf{z}^t$ (Proposition 3.3), aligning with the identification results without instantaneous effect [8, 10, 40, 41]. On the other hand, temporal causal variables with instantaneous effects are shown to be identifiable *only if* "instantaneous parents" (i.e., nodes affecting other nodes instantaneously) are cut by actions [9], reducing to the setting without instantaneous effect where the latent components at $t$ are mutually independent. Upon invariance, more fine-grained latent variable identification results, such as element-wise identifiability, can be obtained by incorporating additional technical assumptions, such as the sparse mechanism shift [8, 41, 43] and parametric latent causal model [40, 46, 47].

**Identification algorithm.** From a high level, the distributional invariance (eq. (D.7)) indicates full explainability and predictability of $\mathbf{z}_{A_t}^t$ from its previous time step $\mathbf{z}^{t-1}$, regardless of the action $\mathbf{a}^t$. In principle, this invariance principle can be enforced by directly maximizing the information content of the proposed default transition density between the learned representation $p(\hat{\mathbf{z}}_{A_t}^t \mid \hat{\mathbf{z}}^{t-1})$ [9, 10]. In practice, the invariance regularization is often incorporated together with the predictability of the variant partition conditioning on actions, implemented as a KL divergence between the observational posterior $q(\hat{\mathbf{z}}^t \mid \mathbf{x}^t)$ and the transitional prior $p(\hat{\mathbf{z}}^t \mid \mathbf{z}^{t-1}, \mathbf{a}^t)$ [8, 11, 39–41, 46], estimated using variational Bayes [48] or normalizing flow [49]. We additionally show that minimizing this KL-divergence $D_{\mathrm{KL}}(q(\hat{\mathbf{z}}^t \mid \mathbf{x}^t) \| p(\hat{\mathbf{z}}^t \mid \mathbf{z}^{t-1}, \mathbf{a}^t))$ is equivalent to maximizing the conditional entropy $p(\hat{\mathbf{z}}_{A_t}^t \mid \hat{\mathbf{z}}^{t-1})$ in App. D.

### D.4 Multi-task Causal Representation Learning

**High-level overview.** Multi-task causal representation learning aims to identify latent causal variables via external supervision, in this case, the label information of the same instance for various tasks. Previously, multi-task learning [50, 51] has been mostly studied outside the scope of identifiability, mainly focusing on domain adaptation and out-of-distribution generalization. One of the popular ideas that was extensively used in the context of multi-task learning is to leverage interactions between different tasks to construct a generalist model that is capable of solving all classification tasks and potentially better generalizes to unseen tasks [52, 53]. Recently, Lachapelle et al. [15], Fumero et al. [16] systematically studied under which conditions the latent variables can be identified in the multi-task scenario and correspondingly provided identification algorithms.

**Data generating process.** The multi-task causal representation learning considers a *supervised* setup: Given a latent SCM as defined in Defn. B.1, we generate the observable $\mathbf{x} \in \mathbb{R}^D$ through some mixing function $f : \mathbb{R}^N \to \mathbb{R}^D$ satisfying Asm. B.1. Given a set of task $\mathcal{T} = \{t_1, \ldots, t_k\}$, and let $\mathbf{y}^k \in \mathcal{Y}_k$ denote the corresponding task label respect to the task $t_k$. Each task only *directly*

depends on a subset of latent variables $S_k \subseteq [N]$, in the sense that the label $\mathbf{y}^k$ can be expressed as a function that contains all and only information about the latent variable $\mathbf{z}_{S_k}$:

$$\mathbf{y}^k = r_k(\mathbf{z}_{S_k}), \tag{D.8}$$

where $r : \mathbb{R}^{|S_k|} \to \mathcal{Y}_k$ is some deterministic function which maps the latent subspace $\mathbb{R}^{|S_k|}$ to the task-specific label space $\mathcal{Y}_k$, which is often assumed to be linear and implemented using a linear readout in practice [15, 16]. For each task $t_k, k \in [K]$, we observe the associated data distribution $P_{\mathbf{x},\mathbf{y}^k}$. Consider two different tasks $t_k, t_{k'}$ with $k, k' \in [K]$, the corresponding data $\mathbf{x}, \mathbf{y}^k$ and $\mathbf{x}, \mathbf{y}^{k'}$ are *invariant* in the intersection of task-related features $\mathbf{z}_A$ with $A = S_k \cap S_{k'}$. Formally, let $r_k^{-1}(\{\mathbf{y}^k\})$ denotes the pre-image of $\mathbf{y}^k$, for which it holds

$$r_k^{-1}(\{\mathbf{y}^k\})_A = r_{k'}^{-1}(\{\mathbf{y}^{k'}\})_A, \tag{D.9}$$

showing alignment on the shared partition of the task-related latents. In the ideal case, each latent component $j \in [N]$ is *uniquely shared* by a subset of tasks, all factors of variation can be fully disentangled, which aligns with the theoretical claims by Lachapelle et al. [15], Fumero et al. [16].

**Identification algorithms.** We remark that the *sharing* mechanism in the context of multi-task learning fundamentally differs from that of multi-view setup, thus resulting in different learning algorithms. Regarding learning, the shared partition of task-related latents is enforced to align up to the linear equivalence class (given a linear readout) instead of sample level $L_2$ alignment. Intuitively, this invariance principle can be interpreted as a soft version of the that in the multiview case. In practice, under the constraint of perfect classification, one employs (1) a sparsity constraint on the linear readout weights to enforce the encoder to allocate the correct task-specific latents and (2) an information-sharing term to encourage reusing latents across various tasks. Equilibrium can be obtained between these two terms only when the shared task-specific latent is element-wise identified (Defn. C.2). Thus, this soft invariance principle is jointly implemented by the sparsity constraint and information sharing regularization [16, Sec. 2.1].

## D.5  Domain Generalization

**High-level overview.** Domain generalization aims at *out-of-distribution* performance. That is, learning an optimal encoder and predictor that performs well at some unseen test domain that preserves the same data symmetries as in the training data. At a high level, domain generalization representation learning [12, 13, 54–56] considers a similar framework as introduced for interventional CRL, with *independent* but *non-identically distributed* data, but additionally incorporated with external supervision and focusing more on model robustness perspective. While interventional CRL aims to identify the true latent factors of variations (up to some transformation), domain generalization learning focuses directly on *out-of-distribution* prediction, relying on some invariance properties preserved under the distributional shifts. Due to the non-causal objective, new methodologies are motivated and tested on real-world benchmarks (e.g., VLCS [57], PACS [58], Office-Home [59], Terra Incognita [60], DomainNet [61]) and could inspire future real-world applicability of causal representation learning approaches.

**Data generating process.** The problem of domain generalizations is an *extension of supervised learning* where training data from multiple environments are available Blanchard et al. [62]. An environment is a dataset of i.i.d. observations from a joint distribution $P_{\mathbf{x}^k,\mathbf{y}^k}$ of the observables $\mathbf{x}^k \in \mathbb{R}^D$ and the label $\mathbf{y}^k \in \mathbb{R}$. The label $\mathbf{y}^k \in \mathbb{R}^m$ only depends on the invariant latents through a linear regression structural equation model [14, Assmp. 1], described as follows:

$$\begin{aligned} \mathbf{y}^k &= \mathbf{w}^* \mathbf{z}_A^k + \epsilon_k, \ \mathbf{z}_A^k \perp \epsilon_k \\ \mathbf{x}^k &= f(\mathbf{z}^k) \end{aligned} \tag{D.10}$$

where $\mathbf{w}^* \in \mathbb{R}^{D \times m}$ represents the ground truth relationship between the label $\mathbf{y}^k$ and the invariant latents $\mathbf{z}_A^k$. $\epsilon_k$ is some white noise with bounded variance and $f : \mathbb{R}^N \to \mathbb{R}^D$ denotes the shared mixing function for all $k \in [K]$ satisfying Asm. B.1. The set of environment distributions $\{P_{\mathbf{x}^k,\mathbf{y}^k}\}_{k \in [K]}$ generally differ from each other because of interventions or other distributional shifts such as covariates shift and concept shift. However, as the relationship between the invariant latents and the labels $\mathbf{w}^*$ and the mixing mechanism $f$ are shared across different environments, the optimal risk remains invariant in the sense that

$$\mathcal{R}_k^*(\mathbf{w}^* \circ f^{-1}) = \mathcal{R}_{k'}^*(\mathbf{w}^* \circ f^{-1}), \tag{D.11}$$

where $\mathbf{w}^*$ denotes the ground truth relation between the invariant latents $\mathbf{z}_A^k$ an the labels $\mathbf{y}^k$ and $f^{-1}$ is the inverse of the diffeomorphism mixing $f$ (see eq. (D.10)). Note that this is a non-trivial $\iota$ property as the labels $\mathbf{y}^k$ only depend on the invariant latents $\mathbf{z}_A^k$, thus satisfying Defn. 2.1 (ii).

**Identification algorithm.** Different distributional invariance are enforced by interpolating and extrapolating across various environments. Among the countless contribution to the literature, *mixup* [54] linearly interpolates observations from different environments as a robust data augmentation procedure, Domain-Adversarial Neural Networks [55] support the main learning task discouraging learning domain-discriminant features, Distributionally Robust Optimization (DRO) [12] replaces the vanilla Empirical Risk objective minimizing only with respect to the worst modeled environment, Invariant Risk Minimization [56] combines the Empirical Risk objective with an invariance constraint on the gradient, and Variance Risk Extrapolation [13, V-REx], similar in spirit combines the empirical risk objective with an invariance constraint using the variance among environments. For a more comprehensive review of domain generalization algorithms, see Zhou et al. [63].

### D.6   Further Explanations for Tab. 2

**General clarification.** Tab. 2 summarizes all special cases of our invariance framework. For each work, we present their technical assumptions, the type of invariance, the implementation for the invariance and the sufficiency regularizers (to satisfy Constraints 3.1 and 3.2), and the type of identifiability they achieve. Note that this table is by no means exhaustive. Also, we omit some additional results and technical assumptions of individual papers for readability. For each line of work, we provide an additional paragraph elaborating on their practical implementation of the invariance principle.

**(a) Single-node intervention and parametric assumptions.** Many existing CRL works that consider single node intervention per node require additional parametric assumptions, either on the mixing function [6, 36] or the latent causal model [5] or both [24], thus achieving (at least) element-wise identifiability (Defn. C.2). Although some proposed algorithms did not directly focus on solving our invariance-based constrained optimization problem (Thm. 3.1) to achieve identifiability, their theoretical identifiability results can be explained using the invariance principle in our framework, as explained in App. D.2.

**(b) Multi-node intervention and linear mixing.** Recently, [25] extends previous interventional CRL works to unknown multi-node interventions and achieves identifiability under the assumption of a linearly independent intervention signature matrix $M_{\text{int}} \in \{0,1\}^{N \times K}$ with each column $k$ represents the intervened node in this environment $k$. The row-wise linear independence of $M_{\text{int}}$ implies that each latent variable must have been intervened at least once. Let $M \in \{0,1\}^{N \times N}$ represent a submatrix of $M_{\text{int}}$ with *linearly independent* columns. By performing the change of basis on $M$ such that only one component is non-zero in each column and projecting the score changes using the corresponding change of basis matrix, the setting becomes similar to the other interventional case (unknown single node intervention per node). This similarity allows it to be intuitively explained using the same distributional invariance principle introduced earlier (App. D.2).

**(c) Paired single-node intervention per node under nonparametric assumptions.** In the nonparametric settings, several works von Kügelgen et al. [2], Varici et al. [7] have shown element-wise latent variable identification under sufficiently different paired perfect intervention per node. By having two sufficiently different interventions per node, one introduces invariance on the interventional target across these paired interventional environments. This invariance property can be enforced using the score differences [7] or algorithmically by performing model selection [2], see App. D.2 for more details.

**(d) Variant latents identification under independence.** While some papers states main identification results on the variant partition, it can be explained by Thm. 3.1 and Proposition 3.3 stating that the variant block can be identified under independence and invertible encoder. For example, Wendong et al. [27, Thm. 4.5] shows block-identifiability on the intervened (variant) latents under [27, Assumption 4.4] of block-wise independence between the invariant and variant blocks.

**(e) Invariance regularizers in temporal CRL.** As explained in App. D.3, instead of directly maximizing the information content of the transition model $H(\mathbf{z}_A^t \mid \mathbf{z}^{t-1})$ on the invariant partition $A$, most temporal CRL minimizes the KL divergence between the observational posterior $q(\mathbf{z}^t \mid \mathbf{x}_t)$ and the transitional prior $p(\mathbf{z}^t \mid \mathbf{z}^{t-1}, \mathbf{a}^t)$ [8, 11, 39, 41, 46]. In the following, we show that

minimizing the KL-divergence $D_{\mathrm{KL}}(q(\mathbf{z}^t \mid \mathbf{x}_t) \parallel p(\mathbf{z}^t \mid \mathbf{z}^{t-1}, \mathbf{a}^t))$ also maximizes the conditional entropy $H(\mathbf{z}_A^t \mid \mathbf{z}^{t-1})$.

First, note that the KL-Divergence can be decomposed given the mutual dependence between the invariant and variant latent partitions:

$$
\begin{aligned}
&D_{\mathrm{KL}}(q(\mathbf{z}^t \mid \mathbf{x}_t) \parallel p(\mathbf{z}^t \mid \mathbf{z}^{t-1}, \mathbf{a}^t)) \\
=&D_{\mathrm{KL}}(q(\mathbf{z}_A^t \mid \mathbf{x}_t) \parallel p(\mathbf{z}_A^t \mid \mathbf{z}^{t-1})) \cdot D_{\mathrm{KL}}(q(\mathbf{z}_{A^{\mathrm{c}}}^t \mid \mathbf{x}_t) \parallel p(\mathbf{z}_{A^{\mathrm{c}}}^t \mid \mathbf{z}^{t-1}, \mathbf{a}^t)),
\end{aligned}
\tag{D.12}
$$

where $A^{\mathrm{c}} : [N] \setminus A$ denotes the variant latent indices. Since KL-divergence is non-negative, the joint KL-divergence is minimized when both additive terms are minimized. Hence, from now on, we focus on the first term $D_{\mathrm{KL}}(q(\mathbf{z}_A^t \mid \mathbf{x}_t) \parallel p(\mathbf{z}_A^t \mid \mathbf{z}^{t-1}))$ where only the invariant partition is involved, which can be rewritten as:

$$
\begin{aligned}
D_{\mathrm{KL}}(q(\mathbf{z}_A^t \mid \mathbf{x}_t) \parallel p(\mathbf{z}_A^t \mid \mathbf{z}^{t-1})) &= \mathbb{E}_{\mathbf{x}_t} \mathbb{E}_{\mathbf{z}_A^t \mid \mathbf{x}_t} \left[ \log \frac{q(\mathbf{z}_A^t \mid \mathbf{x}_t)}{p(\mathbf{z}_A^t \mid \mathbf{z}^{t-1})} \right] \\
=&\mathbb{E}_{\mathbf{x}_t} \left[ H(q(\mathbf{z}_A^t \mid \mathbf{x}_t), p(\mathbf{z}_A^t \mid \mathbf{z}^{t-1})) - H(\mathbf{z}_A^t \mid \mathbf{z}^{t-1}) \right] \\
=&\mathbb{E}_{\mathbf{x}_t} \left[ H(q(\mathbf{z}_A^t \mid \mathbf{x}_t), p(\mathbf{z}_A^t \mid \mathbf{z}^{t-1})) \right] - H(\mathbf{z}_A^t \mid \mathbf{z}^{t-1}).
\end{aligned}
\tag{D.13}
$$

Therefore, minimizing $D_{\mathrm{KL}}(q(\mathbf{z}_A^t \mid \mathbf{x}_t) \parallel p(\mathbf{z}_A^t \mid \mathbf{z}^{t-1}))$ is equivalent to maximizing $H(\mathbf{z}_A^t \mid \mathbf{z}^{t-1})$. Consequently, the commonly used $D_{\mathrm{KL}}(q(\mathbf{z}^t \mid \mathbf{x}_t) \parallel p(\mathbf{z}^t \mid \mathbf{z}^{t-1}, \mathbf{a}^t))$ in the temporal CRL literature is justified as a valid invariance regularizer, enforcing the transitional invariance (eq. (D.7)).

**(f) Invariance regularizers in domain generalization.** While Sagawa et al. [12] directly optimize for the worst-case risk, a link can be drawn between this objective and the risk invariance: Given a pair of linear head $\mathbf{w}$ and encoder $g$ shared across $[K]$ domains, let the order of risks be $\mathcal{R}^{\pi_1} \geq \mathcal{R}^{\pi_2} \dots \mathcal{R}^{\pi_K}$. Since $\mathcal{R}^{\pi_1}$ is lower bounded by $\mathcal{R}^{\pi_2}$ the minimum of the training objective in Sagawa et al. [12] ($\max_{k \in [K]} \mathcal{R}^k(w, g)$) is obtained when $\mathcal{R}^{\pi_1} = \mathcal{R}^{\pi_2}$. Then we have $\mathcal{R}^{\pi_1} = \mathcal{R}^{\pi_2} \geq \dots \geq \mathcal{R}^{\pi_K}$, and the next minimum will be obtained when $\mathcal{R}^{\pi_1} = \mathcal{R}^{\pi_2} = \mathcal{R}^{\pi_3}$, and so on so forth. The optimization procedure stops when the risks are the same across all domains.

[13] minimizes variance between environment risks to enforce the risk invariance, and the we formally show in the following these two are equivalent. Note that the invariance principle for risk alignment can be formulated as:

$$
\mathbb{E}_{k,k'} \left[ \| \mathcal{R}_k - \mathcal{R}_{k'} \|_2^2 \right]
\tag{D.14}
$$

Now we show that minimizing the variance regularizer proposed by Sagawa et al. [12] is equivalent to minimizing the risk alignment term eq. (D.14)

$$
\begin{aligned}
\min \mathrm{Var}\left[\mathcal{R}_k\right] &\equiv \min \mathbb{E}_k \left[ (\mathcal{R}_k)^2 \right] - \mathbb{E}_k \left[ \mathcal{R}_k \right] \\
&\equiv \min \mathbb{E}_{k,k'} \left[ \frac{(\mathcal{R}_k)^2 - 2 \cdot \mathcal{R}_k \cdot \mathcal{R}_{k'} + (\mathcal{R}_{k'})^2}{2} \right] \\
&\equiv \min \mathbb{E}_{k,k'} \left[ (\mathcal{R}_k - \mathcal{R}_{k'})^2 \right] \equiv \min \ eq.\ (D.14)
\end{aligned}
$$

### D.7   Notable Cases Not Directly Covered by the Theory

There are some works are not listed in Tab. 2 that cannot yet be directly explained by our invariance frameworks but are rather loosely connected. One representative line of work [8, 41, 64, 65] relies on the sparsity assumption in the latent dependency to achieve latent variable and graph identification. This assumption is closely related to the *sparse mechanism shift* hypothesis in causal representation learning [1], stating small distributional changes should not affect all causal variables but only a small subset of these. Note that the sparsity constraint is often formulated as the estimator (either for the graph [15, 41] or of the latents [65]) should be at least sparse as the ground truth one, maximizing the cardinality of the unaffected (invariant) part. Some theoretical results do not rely on multiple data pockets that share certain invariance properties but directly employ specific properties within the observational data, such as independent support [3], or shared cluster membership [47, 66].

# E  Proofs

## E.1  Assumption Justification

We justify the Defn. 2.1 (ii) by showing a negative results under violation of the assumption, i.e., trivially invariant latent variables are not identifiable.

**Proposition E.1** (General non-identifiability of trivially invariant latent variables). *Consider the setup in Thm. 3.1, w.l.o.g we assume $\mathfrak{I} = \{\iota\}$ and $\iota$ is trivial in the sense that assumption (ii) in Defn. 2.1 is violated. Then, the corresponding invariant partition $\mathbf{z}_A^k$ is not identifiable for any $k \in [K]$.*

*Proof.* We provide a counter example as follows: Define a trivial $\iota$-property as "if the first component is greater than zero on $A = \{1\}$ of some two dimensional latents $\mathbf{z}$". Formally,

$$\iota(\mathbf{z}_1) = \mathbf{1}[\mathbf{z}_1 > 0].$$

Consider a mixing function $f = id$ and an invertible encoder $g(\mathbf{x}) = g(f(\mathbf{z})) = [\mathbf{z}_1 + \mathbf{z}_2, \mathbf{z}_2]$ satisfying the sufficiency constraint (Constraint 3.2). Define $h_1 = h_2 = [g \circ f]_A$. Then for some realizations $z, \tilde{z}$ with $z_1 + z_2 > 0$ and $\tilde{z}_1 + \tilde{z}_2 > 0$ we have $\iota(h(\mathbf{z})) = \iota(h(\tilde{\mathbf{z}}))$. However, $h_1, h_2$ can not disentangle $\mathbf{z}_1$, showing non-identifiability for the invariant partition $\mathbf{z}_A$. $\qquad\square$

**Link between Defn. 2.1 (ii) and interventional discrepancy.** In the following, we elaborate how Defn. 2.1 (ii) resembles the most common assumption in interventional causal representation learning, the interventional discrepancy [7, 27]. Note that this assumption may termed differently as *sufficient variability* [2, 10], *interventional regularity* [25, 36], but the mathematical formulation remain the same. We begin with restating this assumption:

**Assumption E.1** (Interventional discrepancy [27]). Given $k \in [K]$, let $p_{t_k}$ denote the causal mechanism of the intervened variable $\mathbf{z}_{t_k}$ with $t_k \in [N]$. We say a stochastic intervention $\tilde{p}_k$ satisfies interventional discrepancy if

$$\frac{\partial \log p_{t_k}}{\partial \mathbf{z}_{t_k}}(\mathbf{z}_{t_k} \mid \mathbf{z}_{\mathrm{pa}(t_k)}) \neq \frac{\partial \log \tilde{p}_{t_k}}{\partial \mathbf{z}_{t_k}}(\mathbf{z}_{t_k} \mid \mathbf{z}_{\mathrm{pa}(t_k)}) \qquad \text{almost everywhere } (a.e.).$$

*Proof.* We show that any cases violating the interventional discrepancy assumption also violates Defn. 2.1 (ii) and vice versa. Suppose for a contradiction that there exists $t_k \in [N]$ that is intervened in environment $k \in [K]$, and there is a non-empty interior $U \subset \mathbb{R}$ with non-zero measure where the interventional discrepancy is violated, i.e., for all $z_{t_k} \in U$, it holds

$$\frac{\partial \log p_{t_k}}{\partial z_{t_k}}(\mathbf{z}_{t_k} \mid \mathbf{z}_{\mathrm{pa}(t_k)}) = \frac{\partial \log \tilde{p}_{t_k}}{\partial z_{t_k}}(\mathbf{z}_{t_k} \mid \mathbf{z}_{\mathrm{pa}(t_k)}) \tag{E.1}$$

Note that the invariant partition under a single node imperfect intervention yields the complementary set of the transitive closure of $t_k$, i.e., $A := [N] \setminus \mathrm{TC}(t_k)$ because the (joint) marginal distributional invariance holds in the sense that

$$\iota(\mathbf{z}_A) = p_{\mathbf{z}_A} = \tilde{p}_{\mathbf{z}_A}.$$

W.l.o.g, we assume $A = \{1, \ldots, t_k - 1\}$, define a function $h : \mathbb{R}^N \to \mathbb{R}^{|A|}$ with

$$h(\mathbf{z}) = [\mathbf{z}_1, \ldots, \mathbf{z}_{t_k-2}, \mathbf{z}_{t_k}]$$

that omits the $t_k - 1$ component of $\mathbf{z}$ but includes the variant component $t_k$. Note that the marginal of $\mathbf{z}_{t_k}$ after intervention remains invariant within $U$ because

$$\begin{aligned}
p(\mathbf{z}_{t_k}) &= \int p_{t_k}(\mathbf{z}_{t_k} \mid \mathbf{z}_{\mathrm{pa}(t_k)}) p(\mathbf{z}_{\mathrm{pa}(t_k)}) d\mathbf{z}_{\mathrm{pa}(t_k)} && \mathrm{pa}(t_k) \in A \\
&= \int p_{t_k}(\mathbf{z}_{t_k} \mid \mathbf{z}_{\mathrm{pa}(t_k)}) \tilde{p}(\mathbf{z}_{\mathrm{pa}(t_k)}) d\mathbf{z}_{\mathrm{pa}(t_k)} && eq. \text{ (E.1) and both } p_k, \tilde{p_k} \text{ pdfs} \\
&= \int \tilde{p}_{t_k}(\mathbf{z}_{t_k} \mid \mathbf{z}_{\mathrm{pa}(t_k)}) \tilde{p}(\mathbf{z}_{\mathrm{pa}(t_k)}) d\mathbf{z}_{\mathrm{pa}(t_k)} \\
&= \tilde{p}(\mathbf{z}_{t_k}).
\end{aligned}$$

Therefore, we have $\iota(h(\mathbf{z})) = \iota(h(\tilde{\mathbf{z}}))$ (with $\tilde{\mathbf{z}}$ noting the latent vectors under intervention) contradicting Defn. 2.1 (ii). The other direction (violating Defn. 2.1 (ii) implies violating Asm. E.1) can be proved using the same example.

□

## E.2 Proof for Thm. 3.1

Our proof consists of the following steps:

1. We construct the optimal encoders $G^*$ (Defn. 3.2) and selectors $\Phi^*$ (Defn. 3.4) that solves the constrained optimization problem in Thm. 3.1.

2. We show that, for any invariance property $\iota_i \in \mathfrak{I}$ and any observation $\mathbf{x}^k$ in the corresponding $\iota_i$-equivalent subset $\mathbf{x}_{V_i}$, the selected representation $\phi^{(i,k)} \oslash g_k(\mathbf{x}^k)$ cannot contain any other information than the invariant partition $\mathbf{z}_{A_i}^k$.

3. Lastly, we prove that selected representation $\phi^{(i,k)} \oslash g_k(\mathbf{x}^k)$ relates to the ground truth invariant partition $\mathbf{z}_{A_i}^k$ through a diffeomorphism $h_k : \mathbb{R}^{|A_i|} \to \mathbb{R}^{|A_i|}$ for all invariance property $\iota_i \in \mathfrak{I}$ and for any observable $\mathbf{x}^k$ from the $\iota_i$-equivalent subset $\mathbf{x}_{V_i}$; in other words, $\phi^{(i,k)} \oslash g_k(\mathbf{x}^k)$ block-identifies $\mathbf{z}_{A_i}^k$ in the sense of Defn. 3.1.

**Lemma E.1** (Existence of optimal encoders and selectors). *Consider a set of observables $\mathcal{S}_{\mathbf{x}} = \{\mathbf{x}^1, \mathbf{x}^2, \ldots, \mathbf{x}^K\} \in \mathcal{X}$ generated from § 2 satisfying Asm. 2.1, then there exists optimal encoders $G$ (Defn. 3.2) and selectors $\Phi$ (Defn. 3.4) which satisfy both Constraints 3.1 and 3.2.*

*Proof.* The optimal encoders can be constructed as the set of the inverse of the ground truth mixing functions:

$$G^* = \{f_k^{-1}\}_{k \in [K]}, \tag{E.2}$$

$f_k^{-1}$ is smooth and invertible following Asm. B.1. By definition, for each $k \in [K]$, we have:

$$f_k^{-1}(\mathbf{x}^k) = \mathbf{z}^k \in \mathcal{Z}^k. \tag{E.3}$$

Next, we define the optimal selector $\Phi^* = \{\phi^{(i,k)}\}_{i \in [n_{\mathfrak{I}}], k \in [K]}$ such that for all $i \in n_{\mathfrak{I}}, k \in [K]$, it holds

$$\phi^{(i,k)} \oslash \mathbf{z}^k = \mathbf{z}_{A_i}^k. \tag{E.4}$$

Thus, the invariance constraint (Constraint 3.1) is trivially satisfied as given by § 2. The optimal encoder $f_k^{-1}$ is smooth and invertible following Asm. B.1 so the sufficiency constraint (Constraint 3.2) is also satisfied. Hence, we have shown the existence of the optimum to the constrained optimization problem in Thm. 3.1. □

**Lemma E.2** (Invariant component isolation). *Consider the same set of observables $\mathcal{S}_{\mathbf{x}}$ as introduced in Lemma E.1, then for any set of smooth encoders $G$ (Defn. 3.2), $\Phi$ (Defn. 3.4) that satisfy the invariance condition (Constraint 3.1), the learned representation $\phi^{(i,k)} \oslash g_k(x_k)$ can only be dependent on the invariant latent variables $\mathbf{z}_{A_i}^k := \{\mathbf{z}_i^k : i \in A_i\}$, not any non-invariant variables $\mathbf{z}_j^k$ with $j \in A_i^c := [N] \setminus A_i$.*

*Proof.* This proof directly follows the second assumption (ii) in Defn. 2.1. Define

$$h_k := \phi^{(i,k)} \oslash g_k \circ f_k \quad k \in [K]. \tag{E.5}$$

By Constraint 3.1 and the fact that $f$ and $g$ are diffeomorphisms, we have

$$\iota(h_k(\mathbf{z}^k)) = \iota(h_{k'}(\mathbf{z}^{k'})) \quad a.s. \qquad \forall k < k' \in [K]. \tag{E.6}$$

According to (ii) in Defn. 2.1, $h_k, k \in [K]$ cannot directly depends on any other latent component $\mathbf{z}_q$ with $q \notin A$. Therefore, we have shown that $h_k$ is a function of $\mathbf{z}_{A_i}^k$, for all $k \in [K], \iota_i \in \mathfrak{I}$. □

**Theorem 3.1** (Identifiability of multiple invariant blocks). *Consider a set of observables $\mathcal{S}_{\mathbf{x}} = \{\mathbf{x}^1, \mathbf{x}^2, \ldots, \mathbf{x}^K\}$ with $\mathbf{x}^k \in \mathcal{X}^k$ generated from § 2 satisfying Asm. 2.1. Let $G, \Phi$ be the set of smooth encoders (Defn. 3.2) and selectors (Defn. 3.4) that satisfy Constraints 3.1 and 3.2, then the invariant component $\mathbf{z}_{A_i}^k$ is block-identified (Defn. 3.1) by $\phi^{(i,k)} \oslash g_k$ for all $\iota_i \in \mathfrak{I}, k \in [K]$.*

*Proof.* Lem. E.1 verifies that there exists such optimum which satisfies both invariance and sufficiency conditions (Constraints 3.1 and 3.2). Following Lem. E.2, the composition $\phi^{(i,k)} \oslash g_k$ can only encode information related to the invariant latent subset $A_i$ specified by the invariance property $\iota_i \in \mathfrak{I}$ for all $k \in V_i$. As given by Constraint 3.2, all smooth encoders $g_k \in [K]$ contain at least as much information as the ground truth invariant latents $\mathbf{z}_{A_i}$ for $i$ with $k \in V_i$. Therefore, the selected representation $\phi^{(i,k)} \oslash g_k(\mathbf{x}^k)$ relates to the ground truth invariant partition $\mathbf{z}_{A_i}$ through some diffeomorphism, i.e., $\mathbf{z}_{A_i}$ is blocked-identified by $\phi^{(i,k)} \oslash g_k(\mathbf{x}^k)$ for all invariance property $\iota_i \in \mathfrak{I}$ and observable $k \in V_i$, . $\qquad\square$

### E.3 Proofs for Generalization of Variant Latents

**Proposition 3.2** (General non-identifiability of variant latent variables). *Consider the setup in Thm. 3.1, let $A := \bigcup_{i \in [n_\mathfrak{I}]} A_i$ denote the union of block-identified latent indices and $A^{\mathrm{c}} := [N] \setminus A$ the complementary set where no $\iota$-invariance $\iota \in \mathfrak{I}$ applies, then the variant latents $\mathbf{z}_{A^{\mathrm{c}}}$ cannot be identified.*

*Proof.* We provide a simple counter example with two latent variables $\mathbf{z} = [\mathbf{z}_1, \mathbf{z}_2]$, with the mixing function $f$ being the identity map id. W.l.o.g. we assume the invariant partition to be $A = \{1\}$. According to Thm. 3.1, the invariant latent variable can be identified up to a certain bijection $h : \mathbb{R} \to \mathbb{R}$. Let $\hat{\mathbf{z}}$ be the estimated representation:

$$\hat{\mathbf{z}} = [h(\mathbf{z}_1), \mathbf{z}_2 - \mathbf{z}_1] \tag{E.7}$$

with the estimated mixing function $\hat{f} : \mathbb{R}^2 \to \mathbb{R}^2$:

$$\hat{f}(\hat{\mathbf{z}}) = [h^{-1}(\hat{\mathbf{z}}_1), \hat{\mathbf{z}}_2 + h^{-1}(\hat{\mathbf{z}}_1)], \tag{E.8}$$

then we obtain the same observations $\hat{f}(\hat{\mathbf{z}}) = f(\mathbf{z})$ whereas $\hat{\mathbf{z}}_2$ consists of a mixing of $\mathbf{z}_1$ and $\mathbf{z}_2$, showing the variant latent variable $\mathbf{z}_2$ can not be identified. $\qquad\square$

**Proposition 3.3** (Identifiability of variant latent under independence). *Consider an optimal encoder $g \in G^*$ and optimal selector $\phi \in \Phi^*$ from Thm. 3.1 that jointly identify an invariant block $\mathbf{z}_A$ (we omit subscriptions $k, i$ for simplicity), then $\mathbf{z}_{A^{\mathrm{c}}}(A^{\mathrm{c}} := [N] \setminus A)$ can be identified by the complementary encoding partition $(1 - \phi) \oslash g$ only if: (i) $g$ is invertible in the sense that $I(\mathbf{x}, g(\mathbf{x})) = H(\mathbf{x})$; (ii) $\mathbf{z}_{A^{\mathrm{c}}}$ is independent on $\mathbf{z}_A$.*

*Proof.* Consider the mutual information between the observation $\mathbf{x} \in \mathcal{S}_{\mathbf{x}}$ and the optimal encoder $g \in G^*$ from Thm. 3.1:

$$\begin{aligned} I(\mathbf{x}; g(\mathbf{x})) &= I\left(\mathbf{x}; \phi \oslash g(\mathbf{x}), (1 - \phi) \oslash g(\mathbf{x})\right) \\ &= I\left(\mathbf{x}; \phi \oslash g(\mathbf{x})\right) + I\left(\mathbf{x}; (1 - \phi) \oslash g(\mathbf{x})\right). \end{aligned} \tag{E.9}$$

This decomposition is valid because $\phi \oslash g(\mathbf{x})$ disentangles $\mathbf{z}_A$ from the rest of the encodings, as given by the definition of block-identifiability Defn. 3.1. Therefore, $\phi \oslash g(\mathbf{x})$ is independent on $(1 - \phi) \oslash g(\mathbf{x})$.

Writing $\phi \oslash g(\mathbf{x}) = h(\mathbf{z}_A)$ (Thm. 3.1) and $\mathbf{x} = f(\mathbf{z}_A, \mathbf{z}_{[N] \setminus A})$ with $h : \mathbb{R}^{|A|} \to \mathbb{R}^{|A|}$ some bijection and $f$ the mixing diffeomorphism Defn. B.2, we have:

$$\begin{aligned} I(\mathbf{x}; g(\mathbf{x})) &= I\left(\mathbf{x}; \phi \oslash g(\mathbf{x}), (1 - \phi) \oslash g(\mathbf{x})\right) \\ &= I\left(\mathbf{x}; \phi \oslash g(\mathbf{x})\right) + I\left(\mathbf{x}; (1 - \phi) \oslash g(\mathbf{x})\right) \\ &= I\left(f(\mathbf{z}_A, \mathbf{z}_{A^{\mathrm{c}}}); h(\mathbf{z}_A)\right) + I\left(\mathbf{x}; (1 - \phi) \oslash g(\mathbf{x})\right) \\ &= H(\mathbf{z}_A) + I\left(\mathbf{x}; (1 - \phi) \oslash g(\mathbf{x})\right). \end{aligned} \tag{E.10}$$

Given by condition (i), we have

$$I(\mathbf{x}; g(\mathbf{x})) = H(f(\mathbf{x})) = H(f(\mathbf{z}_A, \mathbf{z}_{A^{\mathrm{c}}})) = H(\mathbf{z}_A) + H(\mathbf{z}_{A^{\mathrm{c}}}), \tag{E.11}$$

cancelling $H(\mathbf{z}_A)$ from both eqs. (E.10) and (E.11), we obtain the following equality:

$$I\left(\mathbf{x}; (1-\phi) \oslash g(\mathbf{x})\right) = H(\mathbf{z}_{A^c}), \tag{E.12}$$

which implies that $(1-\phi) \oslash g(\mathbf{x}) = \tilde{h}(\mathbf{z}_{[N]\setminus A})$ for some bijection $\tilde{h} : \mathbb{R}^{N-|A|} \to \mathbb{R}^{N-|A|}$. That is, the independent complementary block $\mathbf{z}_{A^c}$ is identified by the $(1-\phi) \oslash g(\mathbf{x})$. $\qquad\square$

### E.4 Proofs for Granularity of Latent Variable Identification

**Proposition C.1** (Granularity of identification). *Affine-identifiability (Defn. C.3) implies element-identifiability (Defn. C.2) and block affine-identifiability (Defn. C.1) while element-identifiability and block affine-identifiability implies block-identifiability (Defn. 3.1).*

*Proof.* The diagonal matrix $\Lambda$ in eq. (C.3) is invertible and thus also a diffeomorphism $\phi$ (eq. (C.2)); Diagonal $\Lambda$ of affine identifiability is a special instance of $\tilde{\Lambda}$ in eq. (C.1) where all non-diagonal entries are zero. Hence, affine-identifiability implies element-identifiability and block affine-identifiability. On the other hand, block affine-identifiability is block-identifiability with affine bijection $h$ and element-identifiability defines a special case of block-identifiability where each latent component $\mathbf{z}_i$ is an individual block. $\qquad\square$

**Proposition C.2** (Transition between identification levels). *The transition between different levels of latent variable identification (Fig. 2) can be summarized as follows:*

  (i) *Element-level identifiability (Defns. C.2 and C.3) can be obtained from block-wise identifiability (Defns. 3.1 and C.1) when each individual latent constitutes an invariant block;*

  (ii) *Identifiability up to an affine transformation (Defns. C.1 and C.3) can be obtained from general identifiability on arbitrary diffeomorphism (Defns. 3.1 and C.2) by additionally assuming that both the ground truth mixing function and decoder are finite degree polynomials of the same degree.*

*Proof.* The proof for (i) is trivial in the sense that identification of block with size one boils down to the identification on the element level. The proof for (ii) is based on Ahuja et al. [3, Thm. 4.4] and Zhang et al. [6, Lem. 1], stating that when both ground truth mixing function and decoder are finite degree polynomials of the same degree, the *invertible* encoder learns a representation that is affine linear to the ground truth latents, i.e., $\hat{\mathbf{z}} = \mathbf{L} \cdot \mathbf{z} + \mathbf{b}$ with $\mathbf{L} \in \mathbb{R}^{N \times N}$.

$\qquad\square$

## F  Synthetic Ablation with "Ninterventions"

This subsection presents identifiability results under controversial (non-causal) conditions using simulated data. We consider the synthetic setup with full control over the latent space and the data-generating process. We consider a simple graph of three causal variables as $\mathbf{z}_1 \to \mathbf{z}_2 \to \mathbf{z}_3$. The corresponding joint density has the form of $p_{\mathbf{z}}(z_1, z_2, z_3) = p(z_3 \mid z_2)p(z_2 \mid z_1)p(z_1)$

The goal of this experiment is to demonstrate that existing methods for interventional CRL rely primarily on distributional invariance, regardless of whether this invariance arises from a well-defined intervention or some other arbitrary transformation. To illustrate this, we introduce the concept of a "nintervention," which has a similar distributional effect to a regular intervention, maintaining certain conditionals invariant while altering others, but without a causal interpretation.

**Definition F.1** (Ninterrventions). We define a "*nintervention*" on a causal conditional as the process of changing its distribution but cutting all incoming and outgoing edges. Child nodes condition on the old, pre-intervention, random variable. Formally, we consider the latent SCM as defined in Defn. B.1, an *nintervention* on a node $j \in [N]$ is gives rise to the following conditional factorization $\tilde{p}_{\mathbf{z}}(z) = \tilde{p}(z_j) \prod_{i \in [N]\setminus\{j\}} p(z_i \mid z^{\text{old}}_{\text{pa}(i)})$

Note that the marginal distribution of all non-nintervened nodes $P_{\mathbf{z}_{[N]\setminus j}}$ remain invariant after nintervention. In previous example, we perform a nintervention by replacing the conditional density $p(z_2 \mid z_1)$ using a sufficiently different marginal distribution $p(\tilde{z}_2)$ that satisfies Defn. 2.1 (ii), which gives rise to the following new factorization: $\tilde{p}_{\mathbf{z}}(z_1, z_2, z_3) = p(z_3 \mid z^{\text{old}}_2)\tilde{p}(z_2)p(z_1)$. Note that $\mathbf{z}_3$ conditions on the random variable $\mathbf{z}_2$ before nintervention, whose realization is denoted as $z^{\text{old}}_2$.

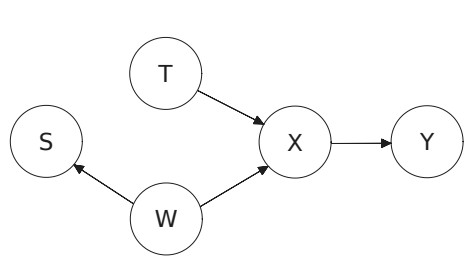

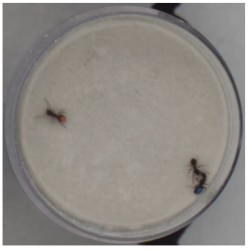

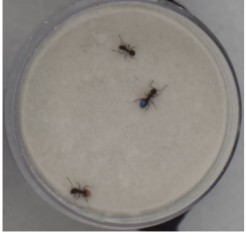

(a) Grooming (blue to focal)   (b) No Action

Figure 3: Causal Model for generic partially annotated scientific experiment: $T$ treatment, $W$ experimental settings, $X$ high-dimensional observation, $Y$ outcome, $S$ annotation flag. Figure and caption adapted from [17, Fig. 1]

Figure 4: Examples of high-dimensional observations $X$ with corresponding annotated social behaviour $Y$ (grooming). Figure and caption adapted from [17, Fig. 2]

Differing from a causal *intervention*, we cut both the incoming and outgoing links of $\mathbf{z}_2$ and keep the marginal distribution of $\mathbf{z}_3$ the same. Clearly, this is a non-sensical intervention from the causal perspective because we eliminates the causal effect from $\mathbf{z}_2$ to its descendants.

**Experiment settings.** As a proof of concept, we choose a linear Gaussian additive noise model and a nonlinear mixing function implemented as a 3-layer invertible MLP with `Leaky-ReLU` activation. We average the results over three independently sampled *ninterventional* densities $\tilde{p}(z_2)$ while guaranteeing all *ninterventional* distributions satisfy Defn. 2.1 (ii). As the marginal distribution of both $\mathbf{z}_1, \mathbf{z}_3$ remains the same after a *nintervention*, we expect $\mathbf{z}_1, \mathbf{z}_3$ to be block-identified (Defn. 3.1) according to Thm. 3.1. In practice, we enforce the marginal invariance constraint (Constraint 3.1) by minimizing the MMD loss, as implemented by the interventional CRL works [6, 37] and train an auto-encoder for a sufficient representation (Constraint 3.2). Further details are included in App. G.

**Results.** To validate block-identifiability, we perform `Kernel-Ridge Regression` between the estimated block $[\hat{\mathbf{z}}_1, \hat{\mathbf{z}}_3]$ and the ground truth latents $\mathbf{z}_1, \mathbf{z}_2, \mathbf{z}_3$ respectively. Both $\mathbf{z}_1, \mathbf{z}_3$ are block-identified, showing a high $R^2$ score of $0.863 \pm 0.031$ and $0.872 \pm 0.035$, respectively. By contrast, the latent variable $\mathbf{z}_2$ is not identified, evidenced by a low $R^2$ of $0.065 \pm 0.017$.

# G Implementation Details

## G.1 Case Study: ISTAnt

**Problem.** Despite the majority of causal representation learning algorithms being designed to enforce the identifiability of some latent factors and tested on controlled synthetic benchmarks, there are a plethora of real-world applications across scientific disciplines requiring representation learning to answer causal questions [67–70]. Recently, Cadei et al. [17] introduced ISTAnt, the first real-world representation learning benchmark with a real causal downstream task (treatment effect estimation). This benchmark highlights different challenges (sources of biases) that could arise from machine learning pipelines even in the simplest possible setting of a randomized controlled trial. Videos of ants triplets are recorded, and a per-frame representation has to be extracted for supervised behavior classification to estimate the Average Treatment Effect of an intervention (exposure to a chemical substance). Beyond desirable identification result on the latent factors (implying that the causal variables are recovered without bias), no clear algorithm has been proposed yet on minimizing the Treatment Effect Bias (TEB) [17]. One of the challenges highlighted by Cadei et al. [17] is that in practice, there is both covariate and concept shifts due to the effect modification from training on a non-random subset of the RCT because, for example, ecologists do not label individual frames but whole video recordings.

**Solution.** Relying on our framework, we can explicitly aim for low TEB by leveraging *known data symmetries* from the experimental protocol. In fact, the causal mechanism ($P(Y^e|do(\mathbf{X}^e = \mathbf{x})$) stays invariant among the different experiment settings (i.e., individual videos or position of the petri dish). This condition can be easily enforced by existing domain generalization algorithms. For exemplary purposes, we choose Variance Risk Extrapolation [13, V-REx], which directly enforces both the invariance sufficiency constraints (Constraints 3.1 and 3.2) by minimizing the Empirical Risk together with the risk variance inter-enviroments.

Table 1: Training setup for synthetic ablations in App. F.

| Parameter | Value |
|---|---|
| Mixing function | 3-layer MLP |
| Encoder | 3-layer MLP |
| Decoder | 3-layer MLP |
| Hidden dim | 128 |
| Activation | Leaky-ReLU |
| Optimizer | Adam |
| Adam: learning rate | 1e-4 |
| Adam: beta1 | 0.9 |
| Adam: beta2 | 0.999 |
| Adam: epsilon | 1e-8 |
| Batch size | 4000 |
| Sample size | 200,000 |
| # Epochs | 500 |

## G.2 Synthetic Ablation with "Ninterventions"

The numerical data is generated using a linear Gaussian additive noise model as follows:

$$
\begin{aligned}
p(\mathbf{z}_1) &= \mathcal{N}(\mu_1, \sigma_1^2) \\
p(\mathbf{z}_2 \mid \mathbf{z}_1) &= \mathcal{N}(\alpha_1 \cdot \mathbf{z}_1 + \beta_1, \sigma_2^2) \\
p(\mathbf{z}_3 \mid \mathbf{z}_2) &= \mathcal{N}(\alpha_2 \cdot \mathbf{z}_2 + \beta_2, \sigma_3^2) \\
\tilde{p}(\mathbf{z}_2) &= \mathcal{N}(\tilde{\mu}_2, \tilde{\sigma}_2^2)
\end{aligned}
\tag{G.1}
$$

We choose $\mu_1 = 10.5, \sigma_1 = 0.8, \alpha_1 = 0.02, \beta_1 = 0, \sigma_2 = 0.5, \alpha_2 = 1, \beta_2 = 3, \sigma_3 = 1, \tilde{\sigma}_2 = 0.02$. We sample three independent $\tilde{\mu}_2$ according to a uniform distribution $\mathrm{Unif}[2, 5]$ to validate the consistency of the identification results.

For the training, we employ a simple auto-encoder architecture implementing both encoder and decoder as 3-Layer MLP. We enforce the marginal invariance using the Max Mean Discrepancy loss (MMD) on the first and last component $\hat{\mathbf{z}}_1, \hat{\mathbf{z}}_3$. Formally, the objective function writes

$$
\mathcal{L}(g, \hat{f}) = \mathbb{E}_{\mathbf{x}, \tilde{\mathbf{x}}} \left[ \left\| \hat{f}(g(\mathbf{x})) - \mathbf{x} \right\|_2^2 + \left\| \hat{f}(g(\tilde{\mathbf{x}})) - \mathbf{x} \right\|_2^2 \right] + \mathrm{MMD}(g(\mathbf{x})_{[1,3]}, g(\tilde{\mathbf{x}})_{[1,3]}),
$$

where $\mathbf{x}, \tilde{\mathbf{x}}$ denote the observational and ninterventional data, respectively.

Further training details are summarized in Tab. 1

## H  Further Discussions and Connections to Other Fields

In this paper, we take a closer look at the wide range of causal representation learning methods. Interestingly, we find that the differences between them may often be more related to "semantics" than to fundamental methodological distinctions. We identified two components involved in identifiability results: preserving information of the data and a set of known invariances. Our results have two immediate implications. First, they provide new insights into the "causal representation learning problem," particularly clarifying the role of causal assumptions. We have shown that while learning the graph requires traditional causal assumptions such as additive noise models or access to interventions, identifying the causal variables may not. This is an important result, as access to causal variables is standalone useful for downstream tasks, e.g., for training robust downstream predictors or even extracting pre-treatment covariates for treatment effect estimation [71], even without knowledge of the full causal graph. Second, we have exemplified how causal representation can lead to successful applications in practice. We moved the goal post from a characterization of specific assumptions that lead to identifiability, which often do not align with real-world data, to a general recipe that allow practitioners to specify known invariances in their problem and learn representations that align with them. In the domain generalization literature, it has been widely observed that invariant training methods often do not consistently outperform empirical risk minimization (ERM). In our experiments, instead, we have demonstrated that the specific invariance enforced by

V-REx [13] entails good performance in our causal downstream task (§ 4). Our paper leaves out certain settings concerning identifiability that may be interesting for future work, such as discrete variables and finite samples guarantees.

One question the reader may ask, then, is "*so what is exactly causal in causal representation learning?*". We have shown that the identifiability results in typical causal representation learning are primarily based on invariance assumptions, which do not necessarily pertain to causality. We hope this insight will broaden the applicability of these methods. At the same time, we used causality as a language describing the "parameterization" of the system in terms of latent causal variables with associated known symmetries. Defining the symmetries at the level of these causal variables gives the identified representation a causal meaning, important when incorporating a graph discovery step or some other causal downstream task like treatment effect estimation. Ultimately, our representations and latent causal models can be "true" in the sense of [72] when they allow us to predict "causal effects that one observes in practice". Overall, our view also aligns with "phenomenological" accounts of causality [73], that define causal variables from a set of elementary interventions. In our setting too, the identified latent variables or blocks thereof are directly defined by the invariances at hand. From the methodological perspective, all is needed to learn causal variables is for the symmetries defined over the causal latent variables to entail some statistical footprint across pockets of data. If variables are available, learning the graph has a rich literature [74], with assumptions that are often compatible with learning the variables themselves. Our general characterization of the variable learning problem opens new frontiers for research in representation learning:

### H.1 Representational Alignment and Platonic Representation

Several works ([75–78]) have highlighted the emergence of similar representations in neural models trained independently. In [78] is hypothesized that neural networks, trained with different objectives on various data and modalities, are converging toward a *shared* statistical model of reality within their representation spaces. To support this hypothesis, they measure the alignment of representations proposing to use a mutual nearest-neighbor metric, which measures the mean intersection of the k-nearest neighbor sets induced by two kernels defined on the two spaces, normalized by k. This metric can be an instance to the distance function in our formulation in Thm. 3.1. Despite not being optimized directly, several models in multiple settings (different objectives, data and modalities) seem to be aligned, hinting at the fact that their individual training objectives may be respecting some unknwon symmetries. A precise formalization of the latent causal model and identifiability in the context of foundational models remains open and will be objective for future research.

### H.2 Environment Discovery

Domain generalization methods generalize to distributions potentially far away from the training, distribution, via learning representations invariant across distinct environments. However this can be costly as it requires to have label information informing on the partition of the data into environments. Automatic environment discovery ([79–81]) attempts to solve this problem by learning to recover the environment partition. This is an interesting new frontier for causal representation learning, discovering data symmetries as opposed to only enforcing them. For example, this would correspond to having access to multiple interventional distributions but without knowing which samples belong to the same interventional or observational distribution. Discovering that a data set is a mixture of distributions, each being a different intervention on the same causal model, could help increase applicability of causal representations to large obeservational data sets. We expect this to be particularly relevant to downstream tasks were biases to certain experimental settings are undesirable, as in our case study on treatment effect estimation from high-dimensional recordings of a randomized controlled trial.

### H.3 Connection with Geometric Deep Learning

Geometric deep learning (GDL) ([82, 83]) is a well estabilished learning paradigm which involves encoding a geometric understanding of data as an inductive bias in deep learning models, in order to obtain more robust models and improve performance. One fundamental direction for these priors is to encode symmetries and invariances to different types of transformations of the input data, e.g. rotations or group actions ([84, 85]), in representational space. Our work can be fundamentally related with this direction, with the difference that we don't aim to model *explicitly* the transformations of the input space, but the invariances defined at the latent level. While an initial connection has been developed for disentanglement [86, 87], a precise connection between GDL and causal

representation learning remains a open direction. We expect this to benefit the two communities in both directions: (i) by injecting geometric priors in order to craft better CRL algorithms and (ii) by incorporating causality into successful GDL frameworks, which have been fundamentally advancing challenging real-world problems, such as protein folding ([88]).

Table 2: **A non-exhaustive summary of existing identifiability results for Causal Representation Learning.** All of the listed works assume injectivity of the mixing function and causal sufficiency (Markovianity) for the causal latent variables. Many listed papers depend on further technical assumptions and could yield additional results. For clarity, these are omitted; see references for details. In the table, "not assigned" means that the practical method did not directly enforce the invariance principle but considered other algorithmic designs that still implicitly preserve the data symmetries..

| Work | Causal Model | Mixing Function | Invariance | Source of invariance, Inv. subset $A$ | Invariance reg. | Sufficiency reg. | Identifiability | Expl. |
|---|---|---|---|---|---|---|---|---|
| Squires et al. [24, Thms. 1 & 2] | linear | linear | distributional | perfect intervention per node | $\mathrm{rank}(H^{\mathsf{T}}\Delta_k H)\overset{!}{=}1$ for source nodes; linear encoder $g(\mathbf{x})=H\mathbf{x}$, where $\Delta_k:=B_k^{\mathsf{T}}B_k-B_0^{\mathsf{T}}B_0, \mathbf{z}=B_k^{-1}\epsilon$ | $g$ invertible by assumption | affine-id. and partial order preserving graph-id. | (a) |
| Ahuja et al. [37, Thm. 2] | nonparam. | finite-deg. poly. | marginal | single-node imperfect interventions on variant latents | $\sum_{k,k'}\sum_{j\in A}$ $\mathrm{MMD}(p_{[g(\mathbf{x})]_j}^k, p_{[g(\mathbf{x})]_j}^{k'})$ | $\sum_k \mathbb{E}_{\mathbf{x}^k}\left\|\hat{f}(g(\mathbf{x}^k))-\mathbf{x}^k\right\|_2^2$ | block affine-id. | - |
| Ahuja et al. [37, Thm. 3] | nonparam. | finite-deg. poly. | marginal | multi-node imperfect interventions on variant latents | $\sum_{k,k'}\sum_{j\in A}$ $\mathrm{MMD}(p_{[g(\mathbf{x})]_j}^k, p_{[g(\mathbf{x})]_j}^{k'})$ | $\sum_k \mathbb{E}_{\mathbf{x}^k}\left\|\hat{f}(g(\mathbf{x}^k))-\mathbf{x}^k\right\|_2^2$ | block affine-id. | - |
| Ahuja et al. [37, Thm. 4] | nonparam. | finite-deg. poly. | marginal support | imperfect interventions on variant latents | $\sum_{k,k'}\sum_{j\in A}$ $\left\|\mathrm{bnd}(\hat{z}_j^k)-\mathrm{bnd}(\hat{z}_j^{k'})\right\|_2^2$ | $\sum_k \mathbb{E}_{\mathbf{x}^k}\left\|\hat{f}(g(\mathbf{x}^k))-\mathbf{x}^k\right\|_2^2$ | block affine-id. | - |
| Buchholz et al. [5] | linear Gaussian | nonparam. | marginal | perfect intervention per node | $-\mathbb{E}_{l\sim\mathcal{U}(\{0,k\})}\mathbb{E}_{\mathbf{x}^l}$ $\ln\left(e^{\mathbf{1}_{l=k}g_k(\mathbf{x}^l)}\right)$ | $\mathbb{E}_{l\sim\mathcal{U}(\{0,k\})}\mathbb{E}_{\mathbf{x}^l}$ $\ln\left(e^{g_k(\mathbf{x}^l)}+1\right)$ | affine id. + graph id. | (a) |

| Work | Causal Model | Mixing Function | Invariance | Source of invariance, Inv. subset $A$ | Invariance reg. | Sufficiency reg. | Identifiability | Expl. |
|---|---|---|---|---|---|---|---|---|
| Varici et al. [36, Thm. 16] | nonparam. | linear | distributional | perfect intervention per node | $\left\|\Delta^s_{\mathbf{x}}(U^{\mathsf{T}})\right\|_0$. For all $j,k\in[N]$, its element $[\Delta^s_{\mathbf{x}}(U^{\mathsf{T}})]_{j,k}=$ $\mathbf{1}([U^{\mathsf{T}}S(\mathbf{x}^0)]_j \overset{P_{\mathbf{x}^0,k}}{\neq} [U^{\mathsf{T}}S(\mathbf{x}^k)]_j),$ $g(\mathbf{x}):=U^+\mathbf{x}$ | $g$ invertible by assumption | affine-id. + graph-id. | (a) |
| Varici et al. [36, Thm. 13] | nonparam. | linear | distributional | imperfect intervention per node | $\left\|\Delta^s_{\mathbf{x}}(U^{\mathsf{T}})\right\|_0$. For all $j,k\in[N]$, its element $[\Delta^s_{\mathbf{x}}(U^{\mathsf{T}})]_{j,k}=$ $\mathbf{1}([U^{\mathsf{T}}S(\mathbf{x}^0)]_j \overset{P_{\mathbf{x}^0,k}}{\neq} [U^{\mathsf{T}}S(\mathbf{x}^k)]_j),$ $g(\mathbf{x}):=U^+\mathbf{x}$ | $g$ invertible by assumption | block affine-id. + graph-id. | (a) |
| Varici et al. [7, Thm. 3] | nonparam. | nonparam. | interventional target | paired perfect intervention per node | $\min\|\Delta^s(g)\|_0$ s.t. it is diagonal. $\Delta^s(g)_{j,k}=$ $\mathbb{E}\left[\left|[S(g(\mathbf{x}^k))-S(g(\mathbf{x}^{k'}))]_j\right|\right]$ | $g$ invertible by assumption | element-id. + graph-id. | (c) |
| Varici et al. [25, Thm. 1] | nonparam. | linear | distributional | linearly independent multi-node perfect intervention | Linear encoder $g(\mathbf{x})=H\mathbf{x}$, $H^*_i \in \mathrm{im}(\Delta s_{\mathbf{x}}\mathbf{w}_i)\backslash\mathrm{span}(H^*_{[i-1]})$ such that the $\dim$ of $\mathrm{proj}_{\mathrm{null}\left(H^*_{[i-1]}\right)}\mathrm{im}(\Delta S_{\mathbf{x}}\mathbf{w}_i)$ equals one. | $g$ invertible by assumption | affine id. + graph id. | (b) |

| Work | Causal Model | Mixing Function | Invariance | Source of invariance, Inv. subset $A$ | Invariance reg. | Sufficiency reg. | Identifiability | Expl. |
|---|---|---|---|---|---|---|---|---|
| Varici et al. [25, Thm. 2] | nonparam. | linear | distributional | linearly independent multinode imperfect intervention | Linear encoder $g(\mathbf{x})=H\mathbf{x}$, $H_i^* \in \mathrm{im}(\Delta s_{\mathbf{x}}\mathbf{w}_i) \backslash \mathrm{span}(H_{[i-1]}^*)$ such that the $\dim$ of $\mathrm{proj}_{\mathrm{null}\left(H_{[i-1]}^*\right)} \mathrm{im}(\Delta S_{\mathbf{x}}\mathbf{w}_i)$ equals one. | $g$ invertible by assumption | block affine-id. + graph id. | (b) |
| Zhang et al. [6] | nonparam. | finite-deg. poly. | distributional | imperfect intervention per node | $-\sum_k \mathrm{MMD}(q_{\tilde{\mathbf{x}}^k}, p_{\mathbf{x}^k})$ where $\tilde{\mathbf{x}}^k$ the generated "counterfactual" pair through VAE | $-\sum_k \mathbb{E}_{\mathbf{x}^k} \log p(\mathbf{x}^k|g(\mathbf{x}^k))$ | affine-id. + graph id. | (a) |
| Wendong et al. [27, Thm. 4.5] | nonparam. | nonparam. | marginal | marginal invariance from multiple fat-hand interventions on the same set of interventional targets $I$, invariant partition $A := [N] \backslash I$ | model selection | $-\sum_k \log p_{\boldsymbol{\theta}}^k(\mathbf{x}^k)$ | block-id. (known graph) | (d) |
| von Kügelgen et al. [2, Thm. 4.1] | nonparam. | nonparam. | interventional target | paired perfect intervention per node | model selection | $-\sum_k \log p_{\boldsymbol{\theta}}^k(\mathbf{x}^k))$ | element-id. + graph-id | (c) |
| von Kügelgen et al. [22] | nonparam. | nonparam. | sample level on all realizations of $z_A^k$ | one imperfect fat-hand intervention | $\left\| g(\mathbf{x}^1)_{\hat{A}} - g(\mathbf{x}^2)_{\hat{A}} \right\|_2$ | $-\sum_k H(g(\mathbf{x}^k)_{\hat{A}}), k \in \{1,2\}$ | block-id. | - |

| Work | Causal Model | Mixing Function | Invariance | Source of invariance, Inv. subset $A$ | Invariance reg. | Sufficiency reg. | Identifiability | Expl. |
|---|---|---|---|---|---|---|---|---|
| Daunhawer et al. [23] | nonparam. | nonparam. | sample level on all realizations of $z_A^k$ | one imperfect fat-hand intervention, | $\left\Vert g_1(\mathbf{x}^1)_{\hat{A}} - g_2(\mathbf{x}^2)_{\hat{A}} \right\Vert_2$ | $-\sum_k H(g_k(\mathbf{x}^k)_{\hat{A}}),$ $k \in \{1,2\}$ | block-id. | - |
| Ahuja et al. [20] | nonparam. | nonparam. | sample level on all realizations of $z_A^k$ | one imperfect fat-hand intervention | $\left\Vert g(\mathbf{x}^1)_{\hat{A}} - g(\mathbf{x}^2)_{\hat{A}} + \delta \right\Vert_2$ | $-\sum_k \mathbb{E}_{\mathbf{x}^k} \log p(\mathbf{x}^k \vert g(\mathbf{x}^k)),$ $k \in \{1,2\}$ | block-id. | - |
| Locatello et al. [19] | nonparam. | nonparam. | sample level | one imperfect fat-hand intervention | avg. encoding | $-\sum_k \mathbb{E}_{\mathbf{x}^k} \log p(\mathbf{x}^k \vert g(\mathbf{x}^k))$ , $k \in \{1,2\}$ | block-id. | - |
| Yao et al. [18, Thm. 3.2] | nonparam. | nonparam. | sample level on all realizations of $z_A^k$ | partial observability | $\sum_{k,k' \in [K]} \left\Vert g_k(\mathbf{x})_{\hat{A}} - g_{k'}(\tilde{\mathbf{x}})_{\hat{A}} \right\Vert_2$ | $-\sum_{k \in [K]} H(g_k(\mathbf{x})_{\hat{A}})$ | block-id. | - |
| Yao et al. [18, Thm. 3.8] | nonparam. | nonparam. | sample level on all realizations of $z_{A_i}^k$ | partial observability, $k \in V_i$ | $\sum_{k,k' \in V_i}$ $\left\Vert g_k(\mathbf{x})_{\hat{A}(i,k)} - g_{k'}(\tilde{\mathbf{x}})_{\hat{A}(i,k')} \right\Vert_2$ | $-\sum_{k \in [K]} H(t_k \circ g_k(\mathbf{x}))$ | block-id | - |
| Brehmer et al. [4] | nonparam. | nonparam. | sample level | perfect intervention per node | $D_{\mathrm{KL}}\big(q(\mathcal{I}, \hat{\mathbf{z}}^{1,2} \mid \mathbf{x}^{1,2}) \Vert p(\mathcal{I}, \hat{\mathbf{z}}^{1,2})\big)$ where $\hat{\mathbf{z}}^k := g(\mathbf{x}^k), k \in \{1,2\}$ | $-\sum_k \mathbb{E}_{\mathbf{x}^k} \log p(\mathbf{x}^k \vert g(\mathbf{x}^k)),$ $k \in \{1,2\}$ | element-id. | - |
| Lippe et al. [10] | nonparam. | nonparam. | transitional invariance on a distributional level | known-target interventions $\mathcal{I}_t$, invariant partition $A := [N] \backslash \mathcal{I}_t$ | $-H(\hat{\mathbf{z}}_{\hat{A}}^t \mid \hat{\mathbf{z}}^{t-1})$ where $\hat{\mathbf{z}}^t := g(\mathbf{x}^t)$ | $-p(\mathbf{x}^t \vert \mathbf{x}^{t-1}, \mathcal{I}_t)$ | block-id. | - |

| Work | Causal Model | Mixing Function | Invariance | Source of invariance, Inv. subset $A$ | Invariance reg. | Sufficiency reg. | Identifiability | Expl. |
|---|---|---|---|---|---|---|---|---|
| Lippe et al. [9] | nonparam. | nonparam. | transitional invariance on a distributional level | known-target, partially perfect interventions $\mathcal{I}_t$, invariant partition $A:=[N]\setminus\mathcal{I}_t$ | $-H(\hat{z}^t_{\hat{A}^t} \mid \hat{z}^{t-1})$ where $\hat{z}^t:=g(x^t)$ | $-p(x^t|x^{t-1},\mathcal{I}_t)$ | block-id. | - |
| Lippe et al. [11] | nonparam. | nonparam. | transitional invariance on a distributional level | binary interventions (interventional target unknown) | $D_{\mathrm{KL}}(q(\hat{z}^t \mid x^t) \| p(\hat{z}^t \mid \hat{z}^{t-1}, r^t))$, $r^t$ observed regime variable | $-\log p(x^t|\hat{z}^t)$ | block-id. | - |
| Lachapelle et al. [15] | nonparam. | nonparam. | task support | task distribution, overlapping task supports, number of causal variables known | $\sum_t \|\hat{w}^{(t)}\|_{2,1}$ | $\sum_t \mathcal{R}(\hat{w}^{(t)} \circ g)$ | affine-id. | (e) |
| Fumero et al. [16] | nonparam. | nonparam. | task support | task distribution, overlapping task supports | $H(\tilde{w})+\sum_t \|\hat{w}^{(t)}\|_1$ | $\sum_t \mathcal{R}(\hat{w}^{(t)} \circ g)$ | element-id. | (e) |
| Sagawa et al. [12] | nonparam. | nonparam. | risk | invariant relationship between label and invariant features, preserved under covariate shift | $\max_{k\in[K]} \mathcal{R}^k(w \circ g)$ | $\max_{k\in[K]} \mathcal{R}^k(w \circ g)$ | NA | (f) |

| Work | Causal Model | Mixing Function | Invariance | Source of invariance, Inv. subset $A$ | Invariance reg. | Sufficiency reg. | Identifiability | Expl. |
|---|---|---|---|---|---|---|---|---|
| Arjovsky et al. [56] | nonparam. | nonparam. | risk | invariant relationship between label and invariant features, preserved under covariate shift | $\left\|\nabla_{\mathbf{w},\mathbf{w}=1}\mathcal{R}^k(\mathbf{w}\circ g)\right\|^2$ | $\sum_{k\in[K]}\mathcal{R}^k(\mathbf{w}\circ g)$ | NA | - |
| Krueger et al. [13] | nonparam. | nonparam. | risk | invariant relationship between label and invariant features, preserved under covariate shift | $\mathrm{Var}(\{\mathcal{R}^k(\mathbf{w}\circ g)\}_{k\in[K]})$ | $\sum_{k\in[K]}\mathcal{R}^k(\mathbf{w}\circ g)$ | NA | (f) |
| Ahuja et al. [14] | nonparam. | nonparam. | risk | invariant relationship between label and invariant features, preserved under covariate shift | $\left\|\nabla_{\mathbf{w},\mathbf{w}=1}\mathcal{R}^k(\mathbf{w}\circ g)\right\|^2$ | $\sum_{k\in[K]}\mathcal{R}^k(\mathbf{w}\circ g)+\mathrm{Var}(\mathcal{R})$ | NA | - |