# OpenReview forum: "Unifying Causal Representation Learning with the Invariance Principle"
_NeurIPS.cc/2024/Workshop/UniReps — UniReps_

### Official Review · Reviewer_wyYB · 2024-10-04
**Interesting theorethical framework**

**Rating:** 7
**Confidence:** 1

**Review:**

Originality, Significance & Theoretical merits:
Overall, this paper provides a novel and unifying view of causal representation learning. Their theory leverages data symmetries rather than relying solely on strict causal assumptions. The authors argue that many existing CRL methods, which claim to guarantee identifiability under certain conditions, can be unified under a broader framework centered around the invariance principle. This perspective highlights the methodological similarity among various CRL approaches and challenges the traditional separation of causal and non-causal assumptions in representation learning.

Unfortunately, this topic is somewhat outside my core area of expertise, and due to time constraints, I was unable to thoroughly review the supplementary materials. As a result, I am not in a position to fully assess the novelty and significance of the proposed theory. I recommend further evaluation by someone with deeper familiarity with causal representation learning and its theoretical foundations.

Technical Soundness:
While the theoretical framework appears to be robust and well-constructed, I am unable to fully assess its validity. A more detailed comparison of this method’s computational requirements—such as time complexity, implementation ease, and performance trade-offs—compared to other approaches, especially when applied to benchmarks like the ISTAnt dataset, would be valuable (for those who are not that familiar with those benchmarks).

Clarity:
The manuscript could benefit from more high-level explanations to provide readers with intuitive insights into how this approach works. Including examples or simplified overviews would help practitioners quickly determine whether the proposed method could be applicable to their specific problems. This would make the core ideas more accessible and actionable, even for those less familiar with the technical details.

---

### Official Review · Reviewer_GaSx · 2024-10-06
**A unifying framework for causal representation learning (CRL) based on the invariance principle.**

**Rating:** 8
**Confidence:** 4

**Review:**

The paper presents a unifying framework for causal representation learning based on the invariance principle. The authors observe that many existing CRL methods, which often rely on specific problem settings and causal assumptions for identifiability, can be unified by focusing on known data symmetries rather than strictly causal relationships.
The use of the invariance principle to align representations with known data symmetries is innovative. The formal definitions and theorems are sound, but the reliance on known invariances requires careful consideration in practice. The proofs provided for latent variable identifiability are significant and contribute to the theoretical foundations of CRL. The distinction between identifying causal variables and causal graphs adds valuable clarity. The application to treatment effect estimation using ecological data demonstrates the practical potential of the framework. However, the experiments would benefit from inclusion of additional datasets to test the framework's robustness across different domains. The paper concludes by emphasizing the shift from strict causal assumptions to preserving data symmetries, which is a meaningful insight that could influence future research in the field.

---

### Official Review · Reviewer_D94w · 2024-10-07

**Rating:** 6
**Confidence:** 3

**Review:**

Summary: The paper introduces a new unified theory for existing nonparametric CRL approaches leveraging the invariance principles and prove latent variable identifiability in this general setting. They show the applicability of their approach to several linear, non-parameteric and linear Gaussian models as seen in Table 2. The paper formalizes the definition of identifiability across many non-parametric causal representation learning papers and shows the application of their framework to certain practical settings with ecology data.

Strengths:
1. I really enjoyed this paper. I think insights that unify theory behind several papers are incredibly useful and this paper provided a large amount of analyses to support their unified framework.
2. While notation heavy, I found the paper pretty easy to follow. I think the high-level overview in section 3 was quite helpful for block identifiability, encoders and selectors, and the constraints.
3. Theorem 3.1 is a strong result showing that given certain constraints, invariant components of the latent variables can be identified.

Weaknesses
1. Complexity: From my understanding of definitions 3.2 - 3.4, the assumption is that you would need to search for encoders and selectors for satisfying constraints. Is this feasible? I think the experimental information is a bit sparse on how simple this would be as well. More examples could be provided on whether this search is truly tractable.
2. Simplified Experimentation: While I appreciate the experiment with the ISTAnt dataset, I found it to be a simplification of the theory and to have some potential issues that should be addressed. First, the paper uses a pretty simple invariance, V-REx. To me, this doesn't represent the full capabilities of the theoretical analysis. Next, the results seem to show incredibly high sensitivity to $\lambda$. This raises questions about the robustness of results.
3. Causal Graph Recovery: Something the entire framework seems to be missing is a discussion on the ability to recover causal structure, which seems to be the main goal of causal representation learning at least to me. I can't tell whether the framework preserves the ability to identify causal relationships between latent variables and instead only focuses on the identification of latent variables. Similarly, I think a "unified theory of causal representation learning" should show an analysis where a causal graph is recovered in Section 4. I might be misunderstanding the goal here but this is my understanding of what a casual model should do.

Overall, I would vote to accept this paper since I think initial results are useful. That being said, I would like more discussion on some of the above in later iterations.

---

### Official Review · Reviewer_WL9o · 2024-10-07

**Rating:** 7
**Confidence:** 3

**Review:**

The paper presents the unifying view of existing causal representation learning works through the lens of the invariance principle. The paper is well-written and the theoretical claims seem sound although I did not check all the math details. Congratulations to the authors for the awesome work!

---

### Decision · Program_Chairs · 2024-10-10

**Decision:**

Accept

**Comment:**

In light of the positive reviewers' feedback and relevancy of the submission, we are pleased to accept this paper for presentation at UniReps 2024. We kindly ask the authors to incorporate the reviewers' suggestions and feedback in the final camera-ready version of the manuscript.